# Universal Cross-Tokenizer Distillation via Approximate Likelihood Matching

**Benjamin Minixhofer** [0x43]    **Ivan Vulić** [0x43]    **Edoardo M. Ponti** [0x45,0x43]

[0x43]University of Cambridge        [0x45]University of Edinburgh

## Abstract

Distillation has shown remarkable success in transferring knowledge from a Large Language Model (LLM) teacher to a student LLM. However, current distillation methods require similar tokenizers between the teacher and the student, restricting their applicability to only a small subset of teacher–student pairs. In this work, we develop a principled cross-tokenizer distillation method to solve this crucial deficiency. Our method is the first to enable effective distillation across fundamentally different tokenizers, while also substantially outperforming prior methods in all other cases. We verify the efficacy of our method on three distinct use cases. First, we show that viewing tokenizer transfer as self-distillation enables unprecedentedly effective transfer across tokenizers, including rapid transfer of subword models to the byte-level. Transferring different models to the same tokenizer also enables ensembling to boost performance. Secondly, we distil a large maths-specialised LLM into a small general-purpose model with a different tokenizer, achieving competitive maths problem-solving performance. Thirdly, we use our method to train state-of-the-art embedding prediction hypernetworks for training-free tokenizer transfer. Our results unlock an expanded range of teacher–student pairs for distillation, enabling new ways to adapt and enhance interaction between LLMs.

## 1   Introduction

Even the latest, most powerful Large Language Models (LLMs) still operate on *tokens*, and therefore require a *tokenizer* — a manually designed component which turns text into a sequence of tokens. Most currently available language models use subword tokenization (Sennrich et al., 2016; Kudo & Richardson, 2018), where each token is a word, or part of a word. Even though subword tokenizers are currently dominant, they are heterogeneously implemented: when a model is released, it often comes with its own tokenizer which has distinct properties such as the actual tokens it comprises (its *vocabulary*) and which method is used to segment text into these tokens (its *tokenization function*). Furthermore, there has been a recent trend away from subwords toward models using vocabularies consisting of characters (Tay et al., 2022; Nawrot et al., 2023) or bytes (e.g. Xue et al., 2022; Yu et al., 2023; Pagnoni et al., 2024). This exemplifies the wildly diverse landscape of tokenization.

On the other hand, distillation (Buciluă et al., 2006; Hinton et al., 2015) has emerged as a powerful paradigm for creating effective language models by training them based on the signal from another model. However, existing distillation methods require similar tokenizers between the teacher and the student. In particular, most prior methods require the teacher and student to represent their input in the same way, i.e., that their tokenizer is equivalent (e.g., Gu et al., 2024; Ko et al., 2024). Some recent methods relax this requirement by developing heuristics to incorporate information from the teacher alongside a main objective (e.g., next-token prediction), making progress toward enabling distillation across similar (but not equivalent) tokenizers (Zhang et al., 2024b; Wan et al., 2024; Boizard et al., 2025). Nonetheless, this requirement still limits their applicability to a rather small subset of the possible teacher–student pairs. In this work, we develop a cross-tokenizer distillation method to solve

39th Conference on Neural Information Processing Systems (NeurIPS 2025).

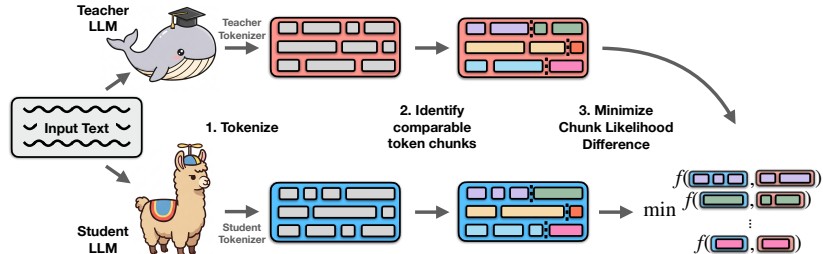

Figure 1: We propose a cross-tokenizer distillation method which identifies comparable chunks of tokens, then minimizes the differences between their likelihoods (c.f. Section 3).

this problem. Our principled objective enables distillation across highly dissimilar tokenizers for the first time (e.g., distilling from a subword tokenizer to a byte-level tokenizer), while also unlocking *pure* distillation — training to purely match the teachers' behaviour, instead of matching the teachers' behaviour as an auxiliary objective, which can lead to further empirical improvements.

In a nutshell, given some text, we first tokenize it with the teacher and the student tokenizers and compute the next-token likelihoods of all tokens under both models. We then find aligned chunks of tokens (the chunks encoding the same portions of text) between the two sequences. Now, to compare the chunk likelihoods of the aligned chunks (e.g., via their KL-divergence), we would need to know the likelihoods of all possible outcomes. This is not an issue in the same-tokenizer distillation case since we can compute the likelihoods of all tokens in the vocabulary in a single forward pass. However, in our case, we would have to compute the likelihoods of all multi-token chunks, which is impossible since their number is infinite. We thus approximate the chunk likelihood difference via a binarised $f$-divergence (Section 3). Together with optional debiasing (Section 3.1) and distillation of the hidden states (Section 3.2), this leads to an effective and universal cross-tokenizer distillation method, which we refer to as Approximate Likelihood Matching (ALM), sketched in Figure 1.

We test ALM on three use cases. First, we propose a novel viewpoint of tokenizer transfer as cross-tokenizer self-distillation (Use Case 1). This enables transferring subword-based LLMs to a different subword tokenizer while largely retaining performance, and achieves rapid and effective Subword $\rightarrow$ Byte transfer for the first time. This allows us to create a competitive byte-level model at a fraction of the cost of training it from scratch. We further show that transferring different models to the same subword tokenizer enables ensembling their predictions (e.g., via averaging their token-level probabilities) which substantially boosts performance over the individual parts. Secondly, we show that ALM outperforms prior methods at distillation of a large math-specialised model into a much smaller general-purpose model with a different tokenizer (Use Case 2). Thirdly, we plug-in ALM as objective to train embedding predictions hypernetworks as introduced by Minixhofer et al. (2024), achieving a new state-of-the-art in tokenizer transfer without ever training on the target tokenizer (i.e., *Zero-Shot* Tokenizer Transfer, Use Case 3). In summary, our work greatly expands the number of possible teacher–student pairs for distillation, improving reusability, composability and transferability of language models. Our code and models are available at `github.com/bminixhofer/tokenkit`.

## 2 Preliminaries and Background

### 2.1 Tokenizers

**Definition.** Tokenizers map the input text to a sequence of tokens. Formally, they operate on finite sequences $\Sigma^\star$ over the alphabet $\Sigma$. For simplicity and since it is not vital to our method, we presume $\Sigma$ to be the set of bytes $\{0, \dots, 255\}$ and $\Sigma^\star$ a string represented as bytes via the UTF-8 standard (Yergeau, 2003). This does not lose generality, since although some tokenizers operate on units built on top of UTF-8 bytes (e.g., Unicode characters; Kudo & Richardson, 2018), they can be transformed to operate on bytes instead (Minixhofer et al., 2024). Following prior work (Uzan et al., 2024; Feher et al., 2024), we denote the tokenizer as the tuple $(\mathcal{V}, T)$ comprising (i) the vocabulary $\mathcal{V}$ which determines the set of all possible tokens and (ii) the tokenization function $T : \Sigma^\star \rightarrow \mathcal{V}^\star$ which determines the sequence of tokens any given text is mapped to.

**Tokenization functions fuse bytes** into larger (e.g., subword) tokens without loss of information.[1] Consider the tokens $\{t_1, t_2, \ldots, t_n\} = T(\boldsymbol{x})$. Then $t_1 \odot t_2 \odot \ldots \odot t_n = \boldsymbol{x}$ (where $\odot$ is the concatenation operator). This has multiple useful implications. All elements in the vocabulary $\mathcal{V}$ are sequences of bytes $\Sigma^{\star}$.[2] Furthermore, $T$ is injective so there exists a left inverse $D$ such that $\boldsymbol{x} = D(T(\boldsymbol{x})) \,\forall \boldsymbol{x}$. We call $D$ the *detokenization function*. Importantly, $D$ is only a *left* inverse, i.e., there may exist a token sequence $\boldsymbol{t}$ such that $\boldsymbol{t} \neq T(D(\boldsymbol{t}))$ since $T$ is not necessarily bijective.

**Tokenization Bias.** Subword tokenizers suffer from what has been termed *tokenization bias* (Phan et al., 2024): sequences of tokens can implicitly leak information about the future content of the text they tokenize. For example, assuming a vocabulary of English words, the token sequence $\{\_\text{Hello}, \_\text{Wor}\}$ leaks that there is no 'ld' after '\_Worl', otherwise the text would have been tokenized as $\{\_\text{Hello}, \_\text{World}\}$ instead (see also Vieira et al., 2024).

## 2.2 Generative Language Models

Generative language models are auto-regressive next-token predictors. Given a sequence of tokens, a generative language model $\theta$ defines a probability distribution $p_\theta(t_i|\boldsymbol{t}_{:i})$[3] over the next token. To compute the probability of a multi-token sequence conditioned on $\boldsymbol{t}_i$, we can compute $p_\theta(\boldsymbol{t}_{i:j}|\boldsymbol{t}_{:i}) = p_\theta(t_i|\boldsymbol{t}_{:i}) \cdot \ldots \cdot p_\theta(t_{j-1}|\boldsymbol{t}_{:j-1})$. To auto-regressively sample from the model, we continuously sample $u \sim p_\theta(u|\boldsymbol{t}_{:i})$ and append the sampled $u$ to the sequence. However, the token level is not an intuitive interface to language models. To obtain a *text* level interface, we need to wrap the language model with tokenization and detokenization. We can compute the probability of a text $\boldsymbol{y}$ conditioned on the text $\boldsymbol{x}$ via $p_\theta(T(\boldsymbol{y})|T(\boldsymbol{x}))$. We can auto-regressively generate a text $\boldsymbol{y}$ conditioned on $\boldsymbol{x}$ by sampling $u_i \sim p_\theta(u_i|T(\boldsymbol{x}) \odot \boldsymbol{u}_{:i})$ for $i \in \{0, \ldots, n\}$, then detokenizing to obtain $\boldsymbol{y} = D(u_0, \ldots, u_n)$. However, for tokenization and detokenization not to introduce any bias, $T(\boldsymbol{x}) \odot T(\boldsymbol{y}) = T(\boldsymbol{x} \odot \boldsymbol{y})$ must hold. This is not always the case due to tokenization bias (see above).

## 2.3 Distillation

**Pure Distillation vs. Hybrid Distillation.** The general idea behind distillation is to transfer knowledge from a teacher model $\phi$ to a student model $\theta$, where the student model has some desirable properties (e.g., being more efficient; Buciluǎ et al., 2006; Hinton et al., 2015).[4] We denote the teacher likelihoods $p_T := p_\phi$ and the student likelihoods $p_S := p_\theta$. A key distinction can be drawn between *pure* distillation, where the primary objective is for the student predictions $p_S(\boldsymbol{y}|\boldsymbol{x})$ to match the teachers' predictions $p_T(\boldsymbol{y}|\boldsymbol{x})$ and *hybrid* distillation, where the main task is to model some ground-truth $p(\boldsymbol{y}|\boldsymbol{x})$, and a distillation objective is used in addition to the main objective (e.g., next-token prediction) to increase performance on the main task. Hybrid distillation can be useful to avoid potential issues arising from the pure setup, e.g. teacher hacking (Tiapkin et al., 2025) and a detrimental student–teacher capacity gap (Busbridge et al., 2025). On the other hand, pure distillation can be highly effective at preserving the student's knowledge when it has the same backbone as the teacher (*self-distillation*); a setting where hybrid distillation can be destructive (c.f. Section 4).

**Distillation Objectives.** To encourage the student predictions to match the teachers', we need some measure of distance between $p_S(\boldsymbol{y}|\boldsymbol{x})$ and $p_T(\boldsymbol{y}|\boldsymbol{x})$. The typical choice is the Kullback-Leibler divergence $D_{\text{KL}}(p_T \parallel p_S) = \sum_{t_i \in \mathcal{V}} p_T(t_i|\boldsymbol{t}_{:i}) \log \frac{p_T(t_i|\boldsymbol{t}_{:i})}{p_S(t_i|\boldsymbol{t}_{:i})}$ (Kullback & Leibler, 1951). However, other objectives have been argued to be more suitable for distillation (Gu et al., 2024; Ko et al., 2024); many of them are instances of $f$-divergences of the form $D_{\text{f}}(p_T \parallel p_S) = \sum_{t_i \in \mathcal{V}} f(p_T(t_i|\boldsymbol{t}_{:i}) \| p_S(t_i|\boldsymbol{t}_{:i}))$ where $f(p_T(x) \| p_S(x)) = p_S(x) g(\frac{p_T(x)}{p_S(x)})$ and $g$ is convex and non-negative (Rényi, 1961).[5]

---

[1]This is not the case for tokenizers which can emit `<unk>` due to out-of-vocabulary characters (such as the original BERT tokenizer; Devlin et al., 2019). However, a tokenizer with this property can again be converted to byte-level and the vocabulary minimally extended to cover all bytes to restore the byte fusion property.

[2]For ease of notation, we do not treat special tokens (such as an `<eos>` token marking the end of the text) separately. Special tokens can be considered a part of the encoding $\Sigma$, e.g., a range from $\{256, \ldots, 256 + k\}$.

[3]We use Python-style indexing where $\boldsymbol{t}_{i:j}$ indicates the elements in $\boldsymbol{t}$ starting from $i$ up to but excluding $j$, and $\boldsymbol{t}_{:j}$ indicates the elements from the beginning (index 0) up to but excluding $j$.

[4]We refer to Xu et al. (2024) for an overview on distillation in the context of LLMs.

[5]Note that while the divergence induced by $f$ is typically defined as $D_f(p\|q) = \sum_x q(x) f(t\frac{p(x)}{q(x)})$ we define it as $D_f(p\|q) = \sum_x f(p(x)\|q(x))$ to retain clarity in absence of the $f$-divergence viewpoint.

**Cross-Tokenizer Distillation.** Recent work has begun investigating ways to distill across different tokenizers, proposing ULD (Boizard et al., 2025), MinED (Wan et al., 2024) and DSKD (Zhang et al., 2024b). We provide a detailed overview of these prior cross-tokenizer distillation methods in Appendix G. They have in common that they heuristically incorporate some information from the teacher. These heuristics are imperfect; prior methods thus strictly operate in the hybrid distillation setup, lacking the properties necessary to form a pure distillation objective (e.g., the loss being minimised iff $p_S(\boldsymbol{y}|\boldsymbol{x}) = p_T(\boldsymbol{y}|\boldsymbol{x}) \; \forall \boldsymbol{x}, \boldsymbol{y}$). Our key contribution is introducing a principled cross-tokenizer distillation objective which enables distillation across fundamentally different tokenizers (e.g., Subword $\rightarrow$ Byte) for the first time, while also unlocking pure-cross tokenizer distillation.

## 3 Methodology

Our key goal is for the student likelihood $p_S(T_S(\boldsymbol{z})|T_S(\boldsymbol{y}))$ to be equal to the teacher likelihood $p_T(T_T(\boldsymbol{z})|T_T(\boldsymbol{y}))$ for every comparable[6] prefix $\boldsymbol{y} \in \Sigma^\star$ and continuation $\boldsymbol{z} \in \Sigma^\star$. To this end, we start by computing the next-token probabilities $p_T(T_T(\boldsymbol{x})_i|T_T(\boldsymbol{x})_{:i})$ and $p_S(T_S(\boldsymbol{x})_i|T_S(\boldsymbol{x})_{:i})$. This can be done in a single forward pass over both language models. We can now enumerate the starting and ending positions of the aligned token chunks between the teacher and the student sequences.[7]

$$A_c(\boldsymbol{x}) = \left\{ (i,j,k,l) \in \mathbb{Z}^4 \;\middle|\; \begin{array}{l} D(T_T(\boldsymbol{x})_{:i}) = D(T_S(\boldsymbol{x})_{:k}) = \boldsymbol{y}, \\ D(T_T(\boldsymbol{x})_{i:j}) = D(T_S(\boldsymbol{x})_{k:l}) = \boldsymbol{z}, \\ c(i,j,k,l) \quad \text{holds} \end{array} \right\} \qquad \text{(Alignment Indices)}$$

Here, $c(i,j,k,l)$ can optionally constrain the alignment to chunks which are amenable to comparison (more on this later in Section 3.1). For now, we define the chunk-level probabilities $p(\boldsymbol{x}, i\!:\!j)$ as

$$p(\boldsymbol{x}, i\!:\!j) \coloneqq p(T(\boldsymbol{x})_{i:j}|T(\boldsymbol{x})_{:i}). \qquad \text{(Chunk-Level Probability)}$$

This lets us define our final objective minimising a divergence between the chunk-level probabilities:

$$\mathcal{L}_{S,T}^{\text{ALM}}(\boldsymbol{x}) = \sum_{i,j,k,l \in A_c(\boldsymbol{x})} f\big(p_T(\boldsymbol{x},i\!:\!j)^{\frac{1}{\tau}} \;||\; p_S(\boldsymbol{x},k\!:\!l)^{\frac{1}{\tau}}\big) + f\big(1 - p_T(\boldsymbol{x},i\!:\!j)^{\frac{1}{\tau}} \;||\; 1 - p_S(\boldsymbol{x},k\!:\!l)^{\frac{1}{\tau}}\big)$$

$$\text{(ALM Objective)}$$

where $\tau$ is a temperature scaling hyperparameter and $f$ is a function inducing an $f$-divergence (Rényi, 1961). Typically, with identical tokenizers, the sum of the $f$-divergence would range over all tokens in the vocabulary; however, when comparing chunk-level probabilities there is an infinite amount of possible outcomes since there is an infinite amount of possible byte sequences, whose probability cannot be estimated in a finite time. We thus resort to computing the $f$-divergence over the binarised possibilities $\{p(\boldsymbol{x}, i:j), 1 - p(\boldsymbol{x}, i:j)\}$ given a sample $\boldsymbol{x} \sim \mathcal{D}$. This constitutes an upper bound to the true $f$-divergence and preserves its crucial properties (e.g., being minimal iff $p_S = p_T$; see Appendix E for further analysis).

### 3.1 Outcome Chunk Debiasing

Our formulation for the Chunk-Level Probability is only approximate since the conditions $T(\boldsymbol{x})_{:i}$ and the outcomes $T(\boldsymbol{x})_{i:j}$ could exhibit different tokenization biases across the teacher and the student. We can avoid the bias stemming from the outcome chunk by modifying the chunk probability to

$$p^{\text{debiased}}(\boldsymbol{x}, i\!:\!j) \coloneqq p(T(\boldsymbol{x})_{i:j}|T(\boldsymbol{x})_{:i}) \cdot \sum \big\{ p(e|T(\boldsymbol{x})_{:j}) \mid e \in \mathcal{V}, e \text{ starts with any } b \in \mathcal{B} \big\}$$

$$\text{(Outcome-Debiased Chunk-Level Probability)}$$

---

[6]Chunks of tokens are strictly only comparable if the difference between their tokenization biases is zero since $p_S(T_S(\boldsymbol{z})|T_S(\boldsymbol{y})) = p_T(T_T(\boldsymbol{z})|T_T(\boldsymbol{y})) \; \forall \boldsymbol{y}, \boldsymbol{z} \in \Sigma^\star$ does *not* imply that the two language models are equivalent: if $T_S(\boldsymbol{z})$ exhibits different tokenization bias from $T_T(\boldsymbol{z})$, the two models implicitly condition on different evidence. We formalize this notion in Appendix F, where we also experiment with explicitly choosing chunks which have sufficiently low differences in tokenization bias.

[7]We compute *greedy* alignments, i.e., we do not consider partially overlapping alignments.

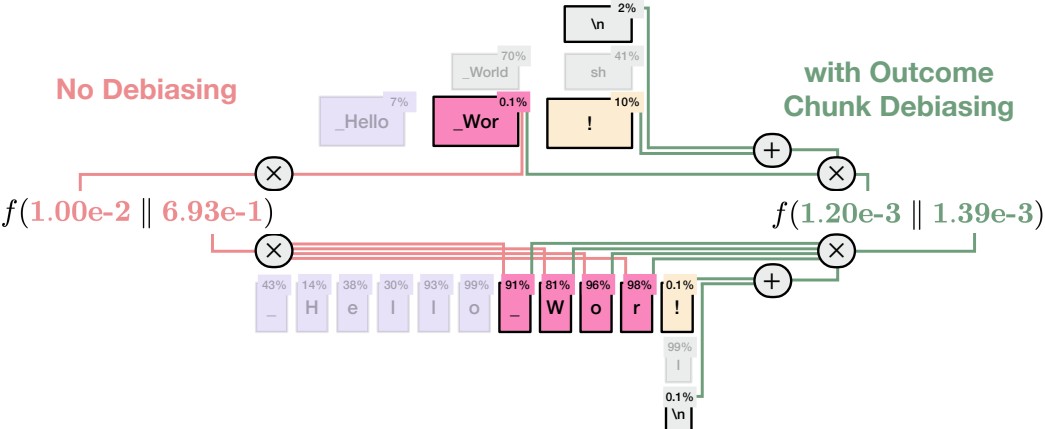

Figure 2: Outcome chunk debiasing removes tokenization bias. For example, the low probability of the subword token `_Wor` would be matched to the high-probability byte sequence $\{$`_`, `W`, `o`, `r`$\}$ in naive subword $\rightarrow$ byte transfer. We can debias by multiplying by the marginal probability of a pretoken-boundary byte occurring after the chunk. In this example, $\{$`\n`, `!`$\} \subseteq \mathcal{B}$ and `s` $\notin \mathcal{B}$, `l` $\notin \mathcal{B}$ where $\mathcal{B}$ is the set of shared pretoken-boundary bytes across the teacher and the student.

Here, $\mathcal{B}$ is the set of pretoken-boundary bytes; these are bytes which never occur within a token. In practice, this can be e.g. whitespace such as the bytes corresponding to the `\n`, `\t` and the space character. Going back to our example from Section 2.1, let $\boldsymbol{t} = \{$`_Hello`, `_Wor`$\}$. By computing $p(\boldsymbol{t}) \cdot \big( p(\{$`_`$\}|\boldsymbol{t}) + (p(\{$`\n`$\}|\boldsymbol{t}) + ...\big)$, we *explicitly* exclude the possibility that $\boldsymbol{t}$ is followed by the letters '`ld`', which, assuming every chosen ending token $e$ starts with a byte in $\mathcal{B}$, removes tokenization bias (otherwise, the sequence would still be biased since there could exist a single token encoding $D(\boldsymbol{t} \odot \{e\})$). This debiasing has been first introduced by Pimentel & Meister (2024) in the context of correctly computing the conditional probability of a word under an autoregressive language model. See Figure 2 for an illustration of this example. Debiasing is only feasibly applicable to the outcome chunks since the required probabilities are computable 'for free' within a single forward pass. There is, however, one remaining complication: if the debiasing probability is very low, the multiplication with $p(T(\boldsymbol{x})_{i:j}|T(\boldsymbol{x})_{:i})$ will 'erase' the chunk probability to an extremely low value, making meaningful comparison between chunk probabilities difficult. To address this issue, we set the alignment constraint (c.f. Equation Alignment Indices):

$$c(i, j, k, l) \coloneqq \sum \big\{ p_T(e|T_T(\boldsymbol{x})_{:j}) \mid e \in \mathcal{V}_T, e \text{ starts with any } b \in \mathcal{B} \big\} \geq \gamma$$

We ablate the impact of outcome chunk debiasing in Appendix A.3, finding that it substantially improves performance, with a slight further improvement by choosing a nonzero threshold $\gamma$.

## 3.2 Distilling Hidden States across Tokenizers

A limitation of our method is that, since we match 32-bit floating point chunk probabilities, the teacher signal has $|\boldsymbol{A_c}| \times 32$ bit per text, while it typically has $\mathcal{O}(|V|)$ more at $|T(\boldsymbol{x})| \times |\mathcal{V}| \times 32$ bit when the teacher and the student have the same tokenizer (i.e., KL divergence between the teacher and the student next-token distributions). To make up for this sparsity of the signal, we optionally add an auxiliary loss maximizing the similarity of aligned hidden states between the two models.[8]

$$\mathcal{L}_{S,T}^{\text{hidden}}(\boldsymbol{x}) = \sum_{l_T, l_S \in L_{T,S}} \sum_{i,j,k,l \in \boldsymbol{A}(\boldsymbol{x})} \| H_T(T_T(\boldsymbol{x}))_j^{l_T} - \text{proj}(H_S(T_S(\boldsymbol{x}))_l^{l_S}) \|$$

(Hidden State Alignment Objective)

where $H_\star(\ldots)_i^l$ are the language models' hidden states at layer $l$ and index $i$ in the sequence and $L_{T,S} \in \mathbb{Z}^2$ are layer-wise alignments between the teacher and the student (e.g., indicating that the last

---

[8]This is inspired by same-tokenizer distillation methods which align hidden states as in Sanh et al. (2020).

Table 1: Results of transferring the Gemma2 2B IT and Llama3.2 3B IT LLMs to the Qwen2 tokenizer and to byte-level tokenization. *original* denotes the original model without transfer. *ARC-C* refers to Arc-Challenge. *AGI-EN* and *AGI-ZH* refer to the English and Chinese splits of AGIEval.

| Model | Tokenizer | Method | Benchmark | | | | | | | Avg. |
| | | | PiQA | ARC-C | BoolQ | MMLU | AGI-EN | AGI-ZH | IFEval | |
|---|---|---|---|---|---|---|---|---|---|---|
| Gemma2 2B IT | | *original* | 79.6 | 50.4 | 83.8 | 56.9 | 42.1 | 30.7 | 62.5 | 58.0 |
| | → Qwen2 | SFT | 75.8 | 43.5 | 77.7 | 49.8 | 31.4 | 28.7 | 54.2 | 51.6 |
| | | DSKD | 74.0 | 41.4 | 79.7 | 50.4 | 33.0 | 28.9 | 53.6 | 51.6 |
| | | MinED | 76.7 | 44.3 | 79.6 | 51.8 | 33.0 | 28.8 | 57.1 | 53.0 |
| | | ALM + SFT | **77.0** | 48.0 | 82.5 | 53.4 | 36.4 | 31.4 | **55.7** | 54.9 |
| | | ALM | 76.8 | **49.0** | **82.7** | **53.6** | **38.9** | **31.6** | 53.2 | **55.1** |
| | → Byte | SFT | 70.7 | 34.7 | 67.9 | 43.1 | 27.6 | 30.0 | 51.5 | 46.5 |
| | | DSKD | 70.5 | 35.6 | 70.2 | 42.3 | 27.5 | **30.4** | **55.0** | 47.4 |
| | | MinED | 69.4 | 35.8 | 72.8 | 42.9 | 28.7 | 29.9 | 50.2 | 47.1 |
| | | ALM + SFT | 71.5 | 38.2 | 80.5 | 51.0 | 35.6 | **30.4** | 51.9 | 51.3 |
| | | ALM | **72.0** | **40.6** | **81.1** | 51.0 | 36.0 | 29.3 | 44.3 | 50.6 |
| Llama3.2 3B IT | | *original* | 76.9 | 43.9 | 78.8 | 62.4 | 36.6 | 40.2 | 76.6 | 59.3 |
| | → Qwen2 | SFT | 76.4 | 44.0 | 80.0 | 60.7 | 33.9 | 29.6 | 65.6 | 55.7 |
| | | DSKD | 72.0 | 37.2 | 78.3 | 45.9 | 32.5 | 30.9 | 60.7 | 51.1 |
| | | MinED | 77.1 | 44.2 | **82.4** | 60.9 | 35.8 | 29.4 | 71.0 | 57.3 |
| | | ALM + SFT | 77.0 | 44.4 | 79.9 | **61.8** | **37.1** | 32.0 | 74.1 | 58.0 |
| | | ALM | **77.3** | **45.6** | 79.0 | 61.6 | **37.1** | **33.3** | **76.3** | **58.6** |
| | → Byte | SFT | **75.2** | 39.8 | 76.8 | 51.5 | 31.5 | 32.6 | **60.8** | 52.6 |
| | | DSKD | 71.1 | 36.0 | 65.8 | 48.0 | 32.0 | 30.3 | 57.9 | 48.7 |
| | | MinED | 73.2 | 38.7 | **78.6** | 51.1 | 33.1 | 32.3 | 59.6 | 52.4 |
| | | ALM + SFT | 73.6 | 39.8 | 76.6 | **57.0** | **35.7** | **33.3** | 58.8 | **53.5** |
| | | ALM | 73.7 | **40.1** | 76.0 | 55.9 | 35.7 | 33.2 | 49.2 | 52.0 |

layer hidden states of the two models should be aligned). $\text{proj}(\star) : \mathbb{R}^{d_S} \to \mathbb{R}^{d_T}$ is a learnt projection function mapping the student hidden states to the teachers' dimensionality. We use greedy alignments (without any constraint $c$) as the hidden state alignments $\boldsymbol{A}(\boldsymbol{x})$. The auxiliary loss enriches the signal from the teacher by $|\boldsymbol{A}| \times d_T \times 32$ bit. We find it to be particularly useful in some cases when the teacher and the student have the same underlying architecture (c.f. Section 4).

## 3.3 Combining Loss Components

As per above, we have two distillation loss components $\mathcal{L}_{S,T}^{\text{ALM}}(\boldsymbol{x})$ and $\mathcal{L}_{S,T}^{\text{hidden}}(\boldsymbol{x})$. We can also combine the distillation loss with a next-token prediction objective (making it *hybrid* distillation; c.f. Section 2.3). The question, then, is how to combine these losses into one? Prior work (Zhang et al., 2024b; Wan et al., 2024, among others) aggregates losses via their arithmetic sum, where the contribution of each loss is optionally modulated by factors $\beta_1, \beta_2, .., \beta_n$. Although this allows choosing $\boldsymbol{\beta}$ such that the losses have approximately equal magnitudes, this does not mean that they contribute equally: they only contribute equally if *their gradients* have equal magnitudes. These two quantities are not necessarily related: for example, forward KL-divergence and cross-entropy have different values but equivalent gradients w.r.t. the predicted probabilities. To solve this problem, GradNorm (Chen et al., 2018) introduces an auxiliary loss to learn task weights such that the per-task gradients $G_W^i$ of the last layer weights $W$ have similar magnitudes across tasks (enumerated by $i$). We take a simpler approach: We compute per-task gradients for the last layer, then set the weight for the task $i$ to $1/\|G_W^i\|$ (normalised to sum to one). Instead of optimizing a loss toward similar last-layer gradient magnitudes, we thus directly solve for equal last-layer gradient magnitudes, choosing only the last layer gradients to maintain minimal overhead. This method (which we refer to as *GradMag*) performs on par or better than GradNorm while being easier to implement (see Appendix A.4).

# 4 Experiments

We experimentally verify our method on three distinct use cases. First, we observe that self-distillation across tokenizers enables unprecedentedly effective transfer of a pre-trained language model to a different tokenizer, which we explore in Use Case 1. Secondly, we find that ALM improves over prior cross-tokenizer distillation objectives at distilling a large maths-specialised model into a much smaller model (Use Case 2). Thirdly, we plug in ALM as a replacement for the next-token prediction objective of Minixhofer et al. (2024)'s hypernetwork training procedure, achieving a new state-of-the-art in Zero-Shot Tokenizer Transfer across a range of tasks (Use Case 3).

**Use Case 1: Tokenizer Transfer via Self-Distillation**

Tokenizer transfer is the problem of transferring a pretrained language model to a different tokenizer which typically has some desirable properties (c.f. Rust et al., 2021). We observe that tokenizer transfer can be treated as cross-tokenizer self-distillation: the teacher is the language model with the original tokenizer, and the student is the same language model with the new tokenizer. We investigate two particular applications of tokenizer transfer via self-distillation. First, we transfer multiple language models to the same (subword) tokenizer; this enables token-level ensembling, improving their performance. Second, we transfer subword-based language models to byte-level tokenization, fundamentally changing their token granularity. This may be an attractive way of creating byte-level models instead of expensively training from scratch.

**Models.** We use the instruction-tuned Gemma2 2B IT (Gemma Team et al., 2024) and the instruction-tuned Llama 3.2 3B IT (Grattafiori et al., 2024) as the models for self-distillation. We transfer these models to the Qwen2 tokenizer; this enables ensembling the models with models of the Qwen series, for which we choose Qwen2.5 1.5B IT (Qwen et al., 2025). We examine the choice of the pivot and the related subtle decision whether to keep or to transfer the special tokens in Appendix H.

**Baseline Methods.** We compare against initialising the new token embeddings using a heuristic (e.g. Tran, 2020; Minixhofer et al., 2022; Gee et al., 2022; Dobler & de Melo, 2023), then training on next-token prediction – the current standard methodology for tokenizer transfer. We refer to this baseline as *SFT*. We use FVT (Gee et al., 2022) as the initialisation heuristic following Minixhofer et al. (2024). We choose DSKD Zhang et al. (2024b) and MinED (Wan et al., 2024) as cross-tokenizer distillation baselines.[9] DSKD and MinED add cross-tokenizer distillation to the next-token prediction objective, so they need to train jointly on SFT and distillation; this is not the case for ALM. We thus also experiment with our objective plus next-token prediction (*ALM + SFT*).

**Training.** We train on the Tulu3 instruction-tuning dataset (Lambert et al., 2025) with LoRA (Hu et al., 2022), sweeping over the learning rate per method (see Appendix A.1) and using GradMag to combine loss components (c.f. Section 3.3); see Appendix B for training details. For transfer to bytes, we make additional adjustments to account for this fundamental change discussed in Appendix C.

**Evaluation & Transfer Results.** We evaluate on a standard set of natural language benchmarks consisting of PiQA (Bisk et al., 2020), ARC-Challenge (Clark et al., 2018), BoolQ (Clark et al., 2019), MMLU (Hendrycks et al., 2021), AGIEval (Zhong et al., 2023)[10] and IFEval (Zhou et al., 2023). We use `lm-eval` (Gao et al., 2024) for all evaluations. Table 1 shows tokenizer transfer results. On transfer to the subword-level Qwen2 tokenizer, MinED leads to consistent improvements over SFT, and ALM and ALM+SFT consistently improve over MinED. Here, pure ALM (without SFT) performs better than ALM+SFT, which may be the case due to better preserving the original model's behaviour. The byte-transfer case is substantially harder: in general, we expect the difficulty of cross-tokenizer distillation to correlate with the difference in vocabulary size between the teacher and the student. The vocabulary size difference is extreme for transfer to bytes. In this case, prior cross-tokenizer distillation methods do not achieve consistent improvements over SFT. In contrast, ALM and ALM+SFT achieve substantial and consistent improvements. We hypothesize ALM+SFT outperforms ALM here since naïve transfer to bytes is necessarily strongly destructive since the model has to adapt to a vastly different granularity, so preserving the original model's behaviour

---

[9]We forgo comparison against ULD (Boizard et al., 2025) since DSKD and MinED have been shown to perform consistently better by Wan et al. (2024) and Zhang et al. (2024b).

[10]For AGIEval, we report the macro-average over the English and Chinese subsets excluding GaoKao Math Cloze and AGIEval Math since they are not designed for instruction-tuned models in their published form.

Table 2: Results of ensembling (transferred) Gemma2 and Llama3 with a Qwen2.5 model. *p* and *logp* denote taking the arithmetic mean of the probabilities and the log probabilities, respectively. *MinED* denotes the ensembling baseline based on heuristically aligning token probabilities.

| | Size | ID | Model | Benchmark | | | | | | | Avg. |
|---|---|---|---|---|---|---|---|---|---|---|---|
| | | | | PiQA | ARC-C | BoolQ | MMLU | AGI-EN | AGI-ZH | IFEval | |
| Original Models | 2.6B | $\mathcal{G}$ | Gemma2 2B IT | 79.6 | 50.4 | 83.8 | 56.9 | 42.1 | 30.7 | 62.5 | 58.0 |
| | 3.2B | $\mathcal{L}$ | Llama3.2 3B IT | 76.9 | 43.9 | 78.8 | 62.4 | 36.6 | 40.2 | 76.6 | 59.3 |
| | 1.5B | $\mathcal{Q}$ | Qwen2.5 1.5B IT | 76.3 | 43.2 | 77.9 | 60.1 | 45.0 | 54.2 | 46.3 | 57.6 |
| Transfers | 2.4B | $\mathcal{G}'$ | $\mathcal{G} \to \mathcal{Q}$ Tok. | 76.8 | 49.0 | 82.7 | 53.6 | 38.9 | 31.6 | 53.2 | 55.1 |
| | 3.3B | $\mathcal{L}'$ | $\mathcal{L} \to \mathcal{Q}$ Tok. | 77.3 | 45.6 | 79.0 | 61.6 | 37.1 | 33.3 | 76.3 | 58.6 |
| Ensembles | 7.3B | | $\mathcal{Q}+\mathcal{G}+\mathcal{L}$ (MinED-logp) | 75.7 | 46.8 | **84.6** | 62.6 | 42.0 | 34.5 | **67.2** | 59.1 |
| | 7.3B | | $\mathcal{Q}+\mathcal{G}+\mathcal{L}$ (MinED-p) | 76.9 | 46.2 | 83.2 | **64.1** | 40.6 | 39.4 | 65.9 | 59.5 |
| | 7.2B | | $\mathcal{Q}+\mathcal{G}'+\mathcal{L}'$ (logp) | **77.7** | **48.4** | 83.9 | 62.2 | **44.2** | **46.8** | 61.1 | 60.6 |
| | 7.2B | | $\mathcal{Q}+\mathcal{G}'+\mathcal{L}'$ (p) | 77.5 | 48.0 | 83.4 | 63.0 | 43.6 | 45.7 | 64.3 | **60.8** |
| Larger Models | 7.6B | | Llama3.1 8B IT | 81.6 | 53.9 | 85.4 | 68.4 | 47.3 | 47.1 | 81.7 | 66.5 |
| | 8.0B | | Qwen2.5 7B IT | 80.4 | 54.3 | 86.4 | 71.7 | 57.6 | 72.6 | 76.6 | 71.4 |

is less crucial. For transfer to bytes, a gap with respect to the original model remains in all cases. There are multiple avenues toward closing this gap: it may be beneficial to add a distillation phase on pretraining data, to add more layers to close the gap in effective parameter count, to retrofit the model into an hourglass architecture with token merging as in DTP (Nawrot et al., 2023) or BLT (Pagnoni et al., 2024), and/or to add multi-byte prediction (Gloeckle et al., 2024).

**Ensembling Results.** Table 2 shows the results of ensembling Qwen with the transferred Gemma and Llama models ($\mathcal{Q} + \mathcal{G}' + \mathcal{L}'$). Following Huang et al. (2024), we compare against a MinED ensemble: this applies the same heuristic as the corresponding distillation method to map tokens in the respective vocabularies to the pivot vocabulary (in this case, Qwen).[11] The ensemble exhibits two notable characteristics: (i) on tasks where all constituents perform at a similar level, the ensemble improves over the performance of each individual model (e.g., PiQA, BoolQ, MMLU) and (ii) on tasks were either one of the constituents performs exceptionally well, the ensemble moves the needle toward this level of performance, although it doesn't reach it (e.g., ARC-C, AGIEval, IFEval). These improvements stem from naïvely averaging the models' predictions. We believe they open the door toward more sophisticated inter-model combination via improved ensembling (e.g., Ormazabal et al., 2023), routing (e.g., Shen et al., 2024) and/or merging (e.g., Sharma et al., 2024).

**Cost of Distillation.** We report the FLOPS and memory requirements as well as the task performance across methods in Figure 3. DSKD requires substantially more TFLOPs than the other methods since there is an extra cross-attention step between the teacher and the student sequences which can incur substantial costs. This is further exacerbated with longer contexts (e.g., when transferring to bytes). MinED needs substantially more memory since the $|T(\boldsymbol{x})| \times |V|$ matrix of logits needs to be aligned with the student along the sequence and vocabulary dimensions. ALM suffers from neither of these constraints and reduces the gap to the teacher by an additional 34% over the best prior method.

### Use Case 2: Large-to-Small Distillation Across Tokenizers

To investigate the efficacy of our method in a large-to-small distillation setting, we scale up from the evaluation setups used in prior work on cross-tokenizer distillation to a realistic larger-scale setup: distilling the maths-specialised OpenMath2-Llama3.1-8B (Toshniwal et al., 2024) into Gemma2 2B. As in Use Case 1, we compare our method against training with SFT, DSKD, and MinED.

**Training.** We use the OpenMathInstruct-2 dataset (which the teacher has been trained on; Toshniwal et al., 2024) and follow the training setup from Use Case 1, apart from increasing the sequence length to 1024 to allow for longer chain-of-thought traces, doubling the batch size to 64, and accordingly reducing the training steps to 5k to train for an equivalent total amount of tokens. We also found it beneficial to train the full model and reduce the learning rate to 5e-6 in preliminary experiments.

---

[11] It is worth noting additional methods to ensemble models with different tokenizers: (i) DeePEn (Huang et al., 2024) and (ii) Pack of LLMs (Mavromatis et al., 2024). The required memory of (i) scales quadratically with the vocabulary size, making it infeasible to apply to models with >100k tokens and (ii) is based on combining the sequence-level likelihoods, which makes generation impossible; we thus compare against MinED.

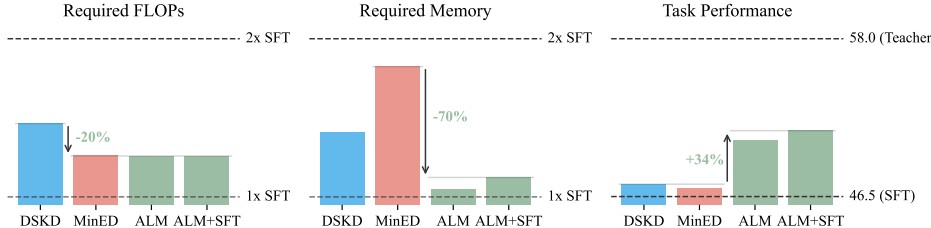

Figure 3: Efficiency and task performance metrics of cross-tokenizer distillation methods, measured via worst-case performance across transfer of Gemma2 to Qwen2 and byte-level tokenizers. *SFT* denotes the required FLOPs and memory as well as the task performance of the SFT baseline.

Table 3: Results of cross-tokenizer distilling the large math-specialized OpenMath2-Llama3.1-8B into the small Gemma2 2B language model. All results are zero-shot CoT.

| Model | Method | GSM8K | MATH | Avg. |
|---|---|---|---|---|
| OpenMath2-Llama3.1-8B | | 88.9 $_{\pm 0.86}$ | 60.2 $_{\pm 0.69}$ | 74.6 $_{\pm 0.55}$ |
| Gemma2 2B IT | | 6.1 $_{\pm 0.66}$ | 11.3 $_{\pm 0.45}$ | 8.7 $_{\pm 0.40}$ |
| | SFT | 67.2 $_{\pm 1.29}$ | 36.2 $_{\pm 0.68}$ | 51.7 $_{\pm 0.73}$ |
| | DSKD | 65.7 $_{\pm 1.31}$ | 34.9 $_{\pm 0.67}$ | 50.3 $_{\pm 0.74}$ |
| Gemma2 2B | MinED | 64.9 $_{\pm 1.31}$ | 34.6 $_{\pm 0.67}$ | 49.8 $_{\pm 0.74}$ |
| | ALM + SFT | **70.2** $_{\pm 1.26}$ | 36.4 $_{\pm 0.68}$ | **53.3** $_{\pm 0.72}$ |
| | ALM | 68.5 $_{\pm 1.28}$ | **36.7** $_{\pm 0.68}$ | 52.6 $_{\pm 0.72}$ |

**Evaluation & Results.** We report zero-shot accuracy on GSM8K (Cobbe et al., 2021) and the MATH benchmark (Hendrycks et al., 2021) in Table 3. Notably, the SFT baseline is competitive, surpassing both MinED and DSKD. However, ALM and ALM + SFT do consistently outperform SFT, leading to an improvement of up to 3% points while greatly surpassing the general-purpose Gemma2 2B IT and achieving 53.3 / 74.6 of the teachers' average performance at 4x fewer parameters.

**Use Case 3: Improving Zero-Shot Tokenizer Transfer Hypernetworks via Self-Distillation**

As a final use case, we apply ALM to Zero-Shot Tokenizer Transfer (ZeTT) hypernetworks (Minix-hofer et al., 2024). In a nutshell, the idea is training a hypernetwork on a distribution of randomly sampled tokenizers such that, after training, the hypernetwork can predict embeddings for any arbitrary tokenizer. The tokenizer transfer effort is thus shifted to a hypernetwork training stage, making subsequent tokenizer transfer easier (or even possible in zero-shot). Minixhofer et al. (2024) train the hypernetworks via a next-token prediction objective, i.e., to predict embeddings which minimize the cross-entropy of the next-token prediction. We instead plug in the ALM loss, training the hypernetwork to produce embeddings which mimic the behaviour of the original model, instead of merely predicting embeddings which perform well at next-token prediction.[12] Results are shown in Table 4. ALM outperforms SFT for hypernetwork training (e.g. 67.5 → 74.0 on BoolQ for zero-shot transfer to the Mistral v2 tokenizer). In general, we expect direct tokenizer transfer self-distillation via ALM as in Use Case 1 to be the most efficient if the goal is transferring to a single target tokenizer. However, if the goal is to potentially transfer to up to $N$ different tokenizers, there is some threshold of $N$ where we expect it would become more efficient to train a ZeTT hypernetwork once, then have a brief additional training phase per tokenizer, instead of separately transferring to the $N$ tokenizers individually. ZeTT hypernetwork training also demonstrates another advantage of our method: while MinED needs an expensive pre-computation step to compute the Levenshtein distance between every pair of tokens, ALM does not require any pre-computation: it can operate in online settings such as hypernetwork training where a new tokenizer is sampled at every training step.

---

[12]We also streamline the hypernetwork architecture to train in a single stage for both methods (Appendix D).

[14]Here, we substitute IFEval for ARC-E (Clark et al., 2018) and HellaSwag (Zellers et al., 2019) to report performance on a superset of the benchmarks used by Minixhofer et al. (2024).

Table 4: Performance of a Gemma2 2B hypernetwork trained with SFT vs. ALM on zero-shot transfer to the GPT2 (Radford et al., 2019), Mistral v2 (Mistral AI, 2025) and Llama3 tokenizers.[14]

| Tokenizer | Method | Benchmark | | | | | | | | Avg. |
|---|---|---|---|---|---|---|---|---|---|---|
| | | PiQA | ARC-E | ARC-C | BoolQ | MMLU | AGI-EN | AGI-ZH | HS | |
| *original* | | 79.6 | 78.2 | 50.4 | 83.8 | 56.9 | 42.1 | 30.7 | 72.6 | 61.8 |
| $\xrightarrow{\text{0-shot}}$ GPT2 | SFT | 74.6 | 63.1 | 40.1 | **73.2** | 40.8 | **26.7** | **27.8** | 68.9 | 51.9 |
| | ALM | **76.1** | **67.2** | **41.4** | 71.6 | **45.0** | 26.1 | 27.2 | **69.7** | **53.0** |
| $\xrightarrow{\text{0-shot}}$ Mistral v2 | SFT | 73.6 | 61.6 | 37.7 | 67.5 | 37.7 | 25.2 | **29.3** | 64.5 | 49.6 |
| | ALM | **75.2** | **66.3** | **40.5** | **74.0** | **38.5** | **26.3** | 28.6 | **66.0** | **51.9** |
| $\xrightarrow{\text{0-shot}}$ Llama3 | SFT | 74.2 | 60.9 | 38.4 | 68.0 | 37.6 | 24.9 | **29.2** | 64.3 | 49.7 |
| | ALM | **75.5** | **66.0** | **40.3** | **74.3** | **38.6** | **25.6** | 28.8 | **65.9** | **51.9** |

Table 5: Results of transferring Gemma3 12B IT to the Qwen2 tokenizer (c.f. Table 1).

| Model | Method | Benchmark | | | | | | | Avg. |
|---|---|---|---|---|---|---|---|---|---|
| | | PiQA | ARC-C | BoolQ | MMLU | AGI-EN | AGI-ZH | IFEval | |
| | *original* | 78.2 | 60.1 | 87.5 | 71.5 | 60.0 | 57.4 | 83.3 | 71.2 |
| Gemma3 12B IT → Qwen2 Tok. SFT | | **81.4** | 56.7 | 83.7 | 67.2 | 46.2 | 43.5 | 73.4 | 64.6 |
| | ALM | 78.8 | **59.9** | **86.9** | **70.7** | **57.6** | **53.6** | **76.7** | **69.2** |

## 4.1 Scaling ALM to Larger Models

We ran additional experiments on tokenizer transfer of Gemma3 12B (Gemma Team et al., 2025) to the Qwen2 tokenizer (also used by Qwen3) via ALM self-distillation to verify that our method scales to larger models. Results are shown in Table 5. Crucially, the transfer-original gap reduction due to ALM scales favourably with model size both in terms of absolute numbers and relative improvements over SFT ($3.5/6.4 \approx 55\%$ reduction for Gemma2 2B → Qwen2 Tok. vs. $4.6/6.6 \approx 70\%$ reduction for Gemma3 12B → Qwen2 Tok.) in this experiment. We leave scaling to truly large models to future work, but note that we expect no fundamental obstacles in doing so (c.f. also Appendix I on an efficient implementation of ALM).

## 5 Conclusion

We have introduced ALM, a principled cross-tokenizer distillation method which outperforms prior heuristic methods across a diverse set of use cases while also enabling distillation across fundamentally different tokenizers for the first time, as evidenced by effective distillation of subword-based models to the byte-level (Use Case 1). We have also shown that ALM is effective at conventional large-to-small distillation (Use Case 2) and enables state-of-the-art zero-shot tokenizer transfer (Use Case 3). Overall, our work vastly expands the number of possible teacher–student pairs for distillation. In addition to enabling a range of new applications, this provides a basis for further enhancements such as more sophisticated combinations of tokenizer-transferred models via enhanced ensembling/routing/merging and using ALM to retrofit subword-based models into dedicated byte-level architectures.

## 6 Limitations

Due to computational constraints, we have trained on comparably few tokens ($\approx 0.6$B). It is not yet clear how ALM compares to SFT and other distillation methods in regimes with larger amounts of training tokens. Furthermore, we only explore one instantiation of cross-tokenizer transfer via minimising a chunk-level $f$-divergence, leaving other instantiations (e.g., a categorical instead of binary divergence) to subsequent explorations. Along the same lines, it is not clear whether our chunk selection strategy is optimal and the space of possible chunk selection strategies is large, including overlapping or non-consecutive chunks. Finally, we have restricted ourselves to the seminal Supervised KD setup (Hinton et al., 2015), leaving applications of ALM to more intricate distillation using e.g. Rejection Sampling (as in Touvron et al., 2023) or On-Policy Distillation (as in Gu et al., 2024; Agarwal et al., 2024) to future work.

## Acknowledgments and Disclosure of Funding

This work is supported by the ERC Starting Grant AToM-FM (101222956) awarded to Edoardo M. Ponti and the Royal Society University Research Fellowship *'Inclusive and Sustainable Language Technology for a Truly Multilingual World'* (no 221137; 2022-) awarded to Ivan Vulić. Research supported by the Google Cloud Research Credits program with the award GCP329647813. Research supported with Cloud TPUs from Google's TPU Research Cloud (TRC). We would like to thank Andreas Grivas, Giwon Hong and Hannah Sterz for their helpful feedback on the paper draft.

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

## A  Ablations & Sensitivity Analyses

### A.1  Sensitivity to the Learning Rate

We analyse the impact of different choices for the learning rate across all methods in Table 6. The different methods are generally robust to settings for the learning rate within the range of $2e-6$ to $5e-5$, although we occasionally observed loss spikes when choosing high learning rates within this range in preliminary experiments. Accordingly, we set the learning rate to $1e-5$ for all methods in our main experiments.

Table 6: Sensitivity of different methods to the learning rate on transfer of Gemma 2 2B IT to the Qwen2 Tokenizer by training for 5k steps (and otherwise matching the experiments in Use Case 1).

| Method | Learning Rate | Benchmark | | | | | | | Avg. |
|---|---|---|---|---|---|---|---|---|---|
| | | PiQA | ARC-C | BoolQ | MMLU | AGI-EN | AGI-ZH | IFEval | |
| SFT | $2e-6$ | 76.1 | 41.8 | **81.1** | **50.4** | **34.5** | **29.9** | 46.8 | 51.5 |
| | $1e-5$ | **76.3** | 42.2 | 80.3 | 50.2 | 33.4 | 29.8 | 51.8 | **52.0** |
| | $5e-5$ | 76.1 | **42.7** | 78.5 | 50.5 | 31.5 | 28.5 | **54.9** | 51.8 |
| DSKD | $2e-6$ | **75.8** | **41.7** | 75.6 | 49.5 | **33.4** | **30.5** | 42.7 | 49.9 |
| | $1e-5$ | 74.6 | 40.9 | 80.0 | 49.0 | 33.1 | 29.8 | 52.2 | 51.4 |
| | $5e-5$ | 74.0 | 41.4 | **80.3** | **51.6** | 32.3 | 29.4 | **54.8** | **52.0** |
| MinED | $2e-6$ | 76.0 | 41.1 | 78.1 | 52.0 | 35.9 | **30.0** | 52.6 | 52.2 |
| | $1e-5$ | **76.6** | **44.0** | **81.9** | **52.2** | **36.3** | 29.7 | 57.5 | **54.0** |
| | $5e-5$ | 76.3 | 43.8 | 80.1 | 51.7 | 32.8 | 28.5 | **58.1** | 53.1 |
| ALM | $2e-6$ | 75.7 | 46.0 | **83.4** | 53.7 | **40.1** | **32.9** | **57.4** | **55.6** |
| | $1e-5$ | **76.8** | 48.0 | 83.1 | **53.8** | 39.3 | 32.0 | 55.5 | 55.5 |
| | $5e-5$ | **76.8** | **49.1** | 81.9 | **53.8** | 38.9 | 30.9 | 54.8 | 55.2 |

### A.2  Impact of the Choice for the Distance Function $f$ and the Temperature $\tau$

The distance function $f$ and the temperature $\tau$ are two crucial hyperparameters of our method. We analyse two choices of $f$, the KL-Divergence $f_{\text{KL}}(p^{1/\tau} \| q^{1/\tau}) = p^{1/\tau} \log \frac{p^{1/\tau}}{q^{1/\tau}}$ and the total variation distance $f_{\text{TVD}}(p^{1/\tau} \| q^{1/\tau}) = |p^{1/\tau} - q^{1/\tau}|$ across a range of temperatures. Figures 4 and 5 show the gradients with respect to $\log q$ over the binarised outcomes of $f_{\text{KL}}$ and $f_{\text{TVD}}$, respectively. For

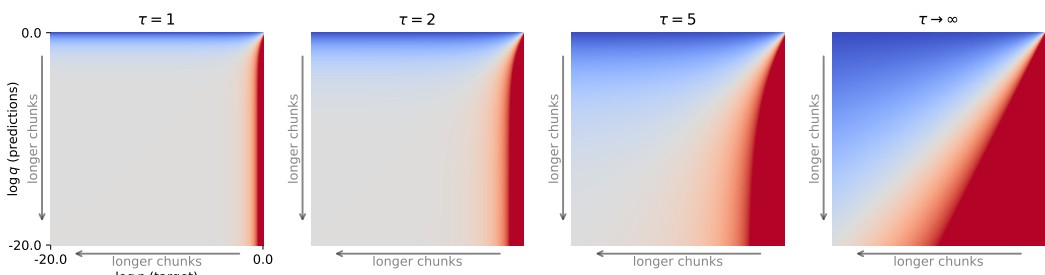

Figure 4: The KL-Divergence gradients $\frac{\delta f_{\text{KL}}(p^{1/\tau\}}\|q^{1/\tau}) + \delta f_{\text{KL}}(1-p^{1/\tau}\|1-q^{1/\tau})}{\delta \log q}$ over $\tau$.

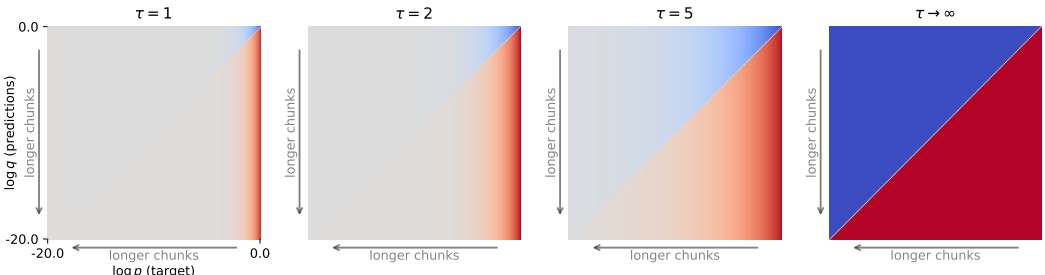

Figure 5: The Total Variation Distance gradients $\frac{\delta f_{\text{TVD}}(p^{1/\tau\}}\|q^{1/\tau}) + \delta f_{\text{TVD}}(1-p^{1/\tau}\|1-q^{1/\tau})}{\delta \log q}$ over $\tau$.

small temperatures (e.g., $\tau = 1$), the gradient vanishes as $\log p$ or $\log q$ decreases. In general, this expected behaviour: mismatches in higher $p$ or $q$ should generally result in a higher contribution to the loss gradients than mismatches when $p$ and $q$ are both low. However, importantly, our method operates on chunks of tokens and the likelihood of a chunk multiplicatively decreases as the length of the chunk increases (since the likelihood of any token is always less than 1); this means that the contribution of longer chunks to the loss vanishes if the temperature $\tau$ is low. The value of $\tau$ thus defines a trade-off: low $\tau$ leads to focusing on high-likelihood chunks. Increasing $\tau$ leads to increased focus on lower-likelihood/longer chunks.[15] We also observe notable behaviour of the KL-Divergence and Total Variation Distance as $\tau \to \infty$, and that the gradients as $\tau \to \infty$ can be computed in closed form. Specifically, $f_{\text{KL}}(p^{1/\tau}\|q^{1/\tau}) + f_{\text{KL}}(1-p^{1/\tau}\|1-q^{1/\tau}) \approx C((\log p - \log q) + \log(p)\log(\frac{\log q}{\log p}))$ and $f_{\text{TVD}}(p^{1/\tau}\|q^{1/\tau}) + f_{\text{TVD}}(1 - p^{1/\tau}\|1 - q^{1/\tau}) \approx C|\log p - \log q|$ as $\tau \to \infty$. To show this, we will use the fact that $\exp(x) \approx 1 + x$ for small enough $x$ and let $a = \log p^{1/\tau} = \log(p)/\tau$ and $b = \log q^{1/\tau} = \log(q)/\tau$, both of which are close to zero since $\tau \to \infty$. We have

$$
\begin{aligned}
f_{\text{KL}}(p^{1/\tau}\|q^{1/\tau}) &= p^{1/\tau} \log \frac{p^{1/\tau}}{q^{1/\tau}} \\
&= \exp(a)\big(\log\exp(a) - \log\exp(b)\big) \\
&= \exp(a)\big(a - b\big) \\
&\approx (1 + a)(a - b) \qquad \text{since } \tau \to \infty \\
&= a + a^2 - b - ab \\
&= \log(p)/\tau + \log(p)^2/\tau^2 - \log(q)/\tau - \log(p)\log(q)/\tau^2 \\
&\approx (\log(p) - \log(q))/\tau \qquad \text{since } 1/\tau \ggg 1/\tau^2 \\
&= C(\log p - \log q) \qquad \text{where the constant } C = 1/\tau
\end{aligned}
$$

and

---

[15]Disentangling the loss contribution by longer chunks compared to shorter lower-likelihood chunks could be an avenue for future work.

$$f_{\text{KL}}(1 - p^{1/\tau} \| 1 - q^{1/\tau}) = (1 - p^{1/\tau}) \log \frac{(1 - p^{1/\tau})}{(1 - q^{1/\tau})}$$

$$= (1 - \exp(a))\big(\log(1 - \exp(a)) - \log(1 - \exp(b))\big)$$

$$\approx -a\big(\log(-a) - \log(-b)\big) \qquad \text{since } \tau \to \infty$$

$$= \log(p)\big(\log(-\log q) - \log(-\log p)\big)/\tau$$

$$= C\left(\log(p) \log\left(\frac{\log q}{\log p}\right)\right) \qquad \text{where the constant } C = 1/\tau$$

Analogously, for the Total Variation Distance we have

$$f_{\text{TVD}}(p^{1/\tau} \| q^{1/\tau}) = f_{\text{TVD}}(1 - p^{1/\tau} \| 1 - q^{1/\tau}) = |p^{1/\tau} - q^{1/\tau}|$$

$$= |\exp(a) - \exp(b)|$$

$$\approx |(1 + a) - (1 + b)| \qquad \text{since } \tau \to \infty$$

$$= |a - b|$$

$$= |\log(p)/\tau - \log(q)/\tau|$$

$$= C|\log(p) - \log(q)| \qquad \text{where the constant } C = 1/\tau$$

Empirically, both KL-divergence and Total Variation Distance perform well, and $\tau > 1$ is beneficial (Table 7). We opt for the KL-divergence for our main experiments and set the temperature $\tau = 100$.[16]

Table 7: Sensitivity to the distance function $f$ and temperature $\tau$ on transfer of Gemma 2 2B IT to the Qwen2 Tokenizer by training for 5k steps (and otherwise matching the experiments in Use Case 1).

| Distance function $f$ | Temperature $\tau$ | Benchmark | | | | | | | Avg. |
|---|---|---|---|---|---|---|---|---|---|
| | | PiQA | ARC-C | BoolQ | MMLU | AGI-EN | AGI-ZH | IFEval | |
| $f = f_{\text{KL}}$ | $\tau = 1$ | 76.4 | 47.2 | 83.0 | 53.6 | **40.0** | 31.4 | **61.1** | 56.1 |
| | $\tau = 5$ | 76.5 | 47.6 | **83.3** | 53.7 | 39.7 | 32.3 | 60.7 | **56.3** |
| | $\tau = 100$ | **76.8** | **48.0** | 83.1 | **53.8** | 39.3 | 32.0 | 55.5 | 55.5 |
| | $\tau \to \infty$ | 76.3 | 47.7 | 83.2 | 53.6 | 39.0 | **32.4** | 54.4 | 55.2 |
| $f = f_{\text{TVD}}$ | $\tau = 1$ | 75.8 | 46.8 | **83.5** | 53.9 | 41.5 | 32.0 | 58.9 | 56.1 |
| | $\tau = 5$ | 75.6 | 47.9 | 83.2 | 53.7 | 40.0 | 32.1 | **61.0** | **56.2** |
| | $\tau = 100$ | **76.3** | **48.9** | 82.9 | 53.6 | 39.2 | 32.2 | 55.6 | 55.5 |
| | $\tau \to \infty$ | 76.1 | 48.4 | 82.9 | 53.7 | 38.6 | **32.9** | 52.9 | 55.1 |

### A.3   Ablating Outcome Chunk Debiasing, the threshold $\gamma$ and Hidden States Distillation

We ablate the impact of Outcome Chunk Debiasing (Section 3.1) as well as the associated threshold and the Hidden State Distillation Loss (Section 3.2) in Table 8. Notably, the hidden state loss is slightly detrimental to Subword $\to$ Subword transfer, but highly beneficial to Subword $\to$ Byte transfer. Outcome Chunk Debiasing is consistently highly beneficial, and can be marginally further improved via a threshold $\gamma > 0$.

### A.4   Comparing Loss Combination Strategies: Fixed $\beta$ vs. GradNorm vs. GradMag

We compare the different loss combination strategies discussed in Section 3.3 in Table 9. GradMag performs on par or slightly better than GradNorm across methods, and both perform vastly better than different fixed combination coefficients $\beta$, while exhibiting only a small overhead from the additional per-task back-propagation through the last layer. Notably, GradNorm requires an extra optimizer (and tuning the associated hyperparameters), which our method does not.

---

[16]$\tau = 5$ performs better and might be a better choice for future experiments. We chose $\tau = 100$ for our main experiments by optimizing for our initial method (see Appendix F); higher temperatures might have been more important for the initial version of our method since it exhibited a longer tail of low-probability chunks.

Table 8: Effect of varying the bias threshold $\gamma$ on transfer of Gemma 2 2B IT to the Qwen2 Tokenizer and Llama 3 3B IT to byte-level tokenization, training for 5k steps and otherwise matching the experiments in Use Case 1. $D$ denotes the addition of Outcome Chunk Debiasing (Section 3.1) and $H$ denotes the addition of the Hidden State Loss (Section 3.1).

| Model → Tokenizer | ALM Setting | Benchmark | | | | | | | Avg. |
|---|---|---|---|---|---|---|---|---|---|
| | | PiQA | ARC-C | BoolQ | MMLU | AGI-EN | AGI-ZH | IFEval | |
| Gemma2 → Qwen2 | $\gamma = 0.10$, D (default) | 76.8 | **48.3** | 83.1 | 53.6 | 38.9 | 32.2 | **56.0** | **55.6** |
| | $\gamma = 0.99$, D | **77.1** | 48.0 | 83.0 | 53.7 | 39.3 | 32.6 | 53.7 | 55.4 |
| | $\gamma = 0.50$, D | 76.6 | 47.9 | **83.4** | 53.5 | **39.8** | **33.2** | 49.6 | 54.9 |
| | $\gamma = 0.00$, D | 76.4 | 47.2 | 83.1 | 53.6 | 39.0 | 31.4 | 58.0 | 55.5 |
| | $\gamma = 0.00$, D, H | 76.6 | 48.0 | 82.8 | **54.0** | 36.6 | 31.5 | 55.8 | 55.0 |
| | $\gamma = 0.00$ | 76.2 | 48.0 | 83.0 | 53.6 | 39.0 | 32.4 | 46.7 | 54.1 |
| Llama3 → Byte | $\gamma = 0.10$, D, H (default) | 73.2 | 38.9 | 77.6 | 55.8 | 34.0 | **32.7** | **18.0** | 47.2 |
| | $\gamma = 0.99$, D, H | 73.1 | 39.8 | **78.2** | 56.1 | **35.3** | 31.8 | 17.3 | **47.4** |
| | $\gamma = 0.50$, D, H | 73.1 | 38.8 | 76.3 | 55.9 | 33.4 | 31.6 | 17.7 | 46.7 |
| | $\gamma = 0.00$, D, H | **73.5** | 38.8 | 77.1 | **56.5** | 34.4 | 30.9 | 17.7 | 47.0 |
| | $\gamma = 0.00$, D | 73.2 | 40.3 | 75.1 | 52.5 | 30.0 | 29.3 | 16.9 | 45.3 |
| | $\gamma = 0.00$ | 73.4 | **41.2** | 74.0 | 52.5 | 30.0 | 31.0 | 0.2 | 43.2 |

Table 9: Analysis of the efficacy of (i) different fixed coefficients $\beta$ (ii) GradNorm (Chen et al., 2018) and (iii) GradMag (Section 3.3) at transfer of Gemma 2 2B IT to the Qwen2 Tokenizer by training for 5k steps and otherwise matching the experiments in Use Case 1.

| Method | Distill. Coef. $\beta$ | Benchmark | | | | | | | Avg. |
|---|---|---|---|---|---|---|---|---|---|
| | | PiQA | ARC-C | BoolQ | MMLU | AGI-EN | AGI-ZH | IFEval | |
| SFT | | 76.3 | 42.2 | 80.3 | 50.2 | 33.4 | 29.8 | 51.8 | 52.0 |
| DSKD | $\beta = 1/3$ | **75.5** | **42.2** | 79.9 | **49.1** | 32.7 | **31.6** | **52.8** | **52.0** |
| | $\beta = 1$ | 75.2 | 39.8 | **80.7** | 47.7 | 32.3 | 30.3 | 49.3 | 50.8 |
| | $\beta = 3$ | 72.5 | 35.5 | 81.3 | 46.6 | 31.6 | 30.9 | 47.9 | 49.5 |
| | GradNorm | 75.3 | 41.6 | 79.7 | 49.0 | 32.8 | 30.3 | 51.3 | 51.4 |
| | GradMag (ours) | 74.6 | 40.9 | 80.0 | 49.0 | **33.1** | 29.8 | 52.2 | 51.4 |
| MinED | $\beta = 1/3$ | 76.5 | 42.5 | 80.4 | 49.8 | 33.4 | 29.4 | 51.8 | 52.0 |
| | $\beta = 1$ | 76.3 | 42.7 | 80.3 | 50.0 | 33.4 | 28.4 | 50.9 | 51.7 |
| | $\beta = 3$ | 76.3 | 43.1 | 80.5 | 50.4 | 33.6 | 29.0 | 51.2 | 52.0 |
| | GradNorm | 76.3 | **44.5** | 81.2 | 50.9 | 36.0 | 29.6 | 53.8 | 53.2 |
| | GradMag (ours) | **76.6** | 44.0 | **81.9** | **52.2** | **36.3** | **29.7** | **57.5** | **54.0** |
| ALM + SFT | $\beta = 1/3$ | 76.3 | 42.3 | 79.9 | 49.8 | 34.0 | 29.3 | 52.0 | 52.0 |
| | $\beta = 1$ | 76.4 | 42.6 | 79.5 | 50.1 | 34.4 | 29.4 | 52.2 | 52.1 |
| | $\beta = 3$ | 76.3 | 42.2 | 80.0 | 50.4 | 34.7 | 30.0 | 52.6 | 52.3 |
| | GradNorm | **77.1** | **46.0** | **83.2** | **53.7** | **38.7** | **32.2** | **61.6** | **56.1** |
| | GradMag (ours) | 76.7 | **46.0** | 82.8 | 53.4 | 37.9 | 32.1 | 58.8 | 55.4 |

## B  Training Details

Across experiments, unless specified otherwise, we use a batch size of 64 texts and a sequence length of 512 tokens for both student and teacher. We use Adam (Kingma & Ba, 2015) without weight decay following the adaptations settings of Groeneveld et al. (2024). We choose a default peak learning rate of $1e - 5$ based on the findings in Appendix A.1, training for 20k steps with linear warmup over 2k steps, then linear decay to zero. We use GradMag (c.f. Section 3.3) to balance loss components. We use LoRA (Hu et al., 2022) with $\alpha = r = 64$. For ALM, we use a threshold $\gamma = 0.1$, the temperature $\tau = 100$, and set $f$ to $f_{\mathrm{KL}}(p_T \| p_S) = p_T \log \frac{p_T}{p_S}$ to recover the KL-divergence, analysing the impact of these choices in Appendix A.2. We conduct all experiments on a cluster of 40 v3 TPU chips and 64 v4 TPU chips. The largest individual experiments run on a pod of 32 v4 TPU chips and take $\approx 24$ hours for transfer of Llama3 to byte-level tokenization, $\approx 12$ hours for transfer of Gemma2 to byte-level tokenization, $\approx 2$ days for Gemma2 hypernetwork training and $\approx 5$ to 10 hours individually per remaining experiment.

# C  Adjustments for Transfer to Bytes in Use Case 1

For the transfer to bytes, we start from the hyperparameters described in Use Case 1, with the following modifications. We quadruple the student sequence length to 2048,[17] halve the batch size to 32 texts, increase the learning rate to 3e-5 and train the full model. Adding the hidden state distillation loss leads to consistent improvements (while results are mixed in the subword case, c.f. Appendix A.3), so we add it here and do not add it for subword-to-subword transfer. We also untie the embedding matrices since sharing the input and output embedding parameters has negligible impact on the parameter count when the vocabulary just consists of the 255 possible bytes. Transfer to bytes substantially decreases the parameter count due to shrinking the embedding matrix (which can make up a considerable portion of the total parameters). Thus, to make it less destructive, we devise a strategy to re-incorporate the subword embedding parameters: for every byte position, we backtrack to find the longest matching subword token ending at this position. We then add the embedding of this token to the byte embedding. This is akin to BLT's strategy of adding n-gram embeddings Pagnoni et al. (2024), where in our case the vocabulary is given by the subword-based model. This strategy only accounts for the parameter mismatch at the *input layer*, not at the output layer, so we might still expect decreased performance.

# D  Hypernetwork Architecture Modifications for Use Case 3

We simplify the hypernetwork training procedure introduced by Minixhofer et al. (2024) to only require a single stage instead of the original two-stage training procedure and remove the necessity of the auxiliary embedding similarity loss. While Minixhofer et al. (2024) predict the target embeddings as $E^{\text{out}} = H_\theta(E^{\text{in}})$, we switch to a residual formulation $E^{\text{out}} = E^{\text{in}}[0] + H_\theta(E^{\text{in}})$ where $H_\theta$ is the hypernetwork and $E^{\text{in}}$ is the input embedding sequence. This way, the hypernetwork (i) does not need to initially learn to mimic the input embeddings for sequences of length 1 (the purpose of the original stage one) and (ii) is less prone to catastrophically drifting in embedding space throughout training (enabling removing the auxiliary loss). We first reproduce Minixhofer et al. (2024)'s results on the TinyLlama model (Zhang et al., 2024a), then test the impact of switching to our residual formulation in Table 10. Based on these findings, we use a residual hypernetwork without auxiliary loss for our main experiments.

Table 10: Performance of a TinyLlama hypernetwork trained with Minixhofer et al. (2024) setting, vs. our residual setting. While removing the auxiliary loss is catastrophic in Minixhofer et al. (2024)'s setting, our residual setting allows removing it at only a minor performance degradation. We exclude BoolQ and MMLU from this comparison since TinyLlama performance is not better than random.

| Method | Tokenizer | Benchmark | | | | | | Avg. |
|---|---|---|---|---|---|---|---|---|
| | | PiQA | ARC-E | ARC-C | AGI-EN | AGI-ZH | HS | |
| *original* | | 73.1 | 55.2 | 30.7 | 23.3 | 27.8 | 59.1 | 44.9 |
| Minixhofer et al. (2024) (reproduced) | $\xrightarrow{\text{0-shot}}$ GPT2 | 70.9 | 48.8 | 28.3 | 22.9 | 27.5 | 55.9 | 42.4 |
| | $\xrightarrow{\text{0-shot}}$ Mistral v2 | 70.6 | 50.3 | 28.8 | 21.9 | 27.9 | 54.0 | 42.2 |
| | $\xrightarrow{\text{0-shot}}$ Llama3 | 70.5 | 49.7 | 28.2 | 21.9 | 27.1 | 53.8 | 41.9 |
| | Average | 70.6 | 49.6 | 28.4 | **22.2** | **27.5** | 54.5 | 42.2 |
| SFT (Residual + Lex.) | $\xrightarrow{\text{0-shot}}$ GPT2 | 71.4 | 50.1 | 30.4 | 21.5 | 26.6 | 56.4 | 42.7 |
| | $\xrightarrow{\text{0-shot}}$ Mistral v2 | 70.3 | 50.3 | 29.7 | 21.2 | 27.1 | 54.7 | 42.2 |
| | $\xrightarrow{\text{0-shot}}$ Llama3 | 71.0 | 49.8 | 29.6 | 20.8 | 26.5 | 54.2 | 42.0 |
| | Average | **70.9** | **50.1** | **29.9** | 21.2 | 26.7 | **55.1** | **42.3** |
| SFT (Residual) | $\xrightarrow{\text{0-shot}}$ GPT2 | 70.4 | 49.2 | 28.9 | 21.8 | 26.2 | 55.9 | 42.1 |
| | $\xrightarrow{\text{0-shot}}$ Mistral v2 | 69.4 | 49.6 | 29.1 | 21.5 | 26.8 | 53.8 | 41.7 |
| | $\xrightarrow{\text{0-shot}}$ Llama3 | 70.0 | 49.5 | 28.5 | 22.1 | 26.5 | 53.5 | 41.7 |
| | Average | 69.9 | 49.4 | 28.8 | 21.8 | 26.5 | 54.4 | 41.8 |

---

[17]The increase in sequence length is necessary since one token consists of $\approx 4$ bytes on average.

# E  Relation of the Binarised $f$-divergence to the Categorical $f$-divergence

Recall that we define the divergence induced by $f$ as $D_f(p\|q) = \sum_{x \sim D} f\big(p(x)\|q(x)\big)$. Here, $f$ must be writable as $f(p(x)\|q(x)) = q(x)g\left(\frac{p(x)}{q(x)}\right)$ where $g$ is convex and non-negative by the definition of $f$-divergence (Rényi, 1961). The binarised $f$-divergence is $D_f^{\text{binarised}}(p\|q) = f(p(x)\|q(x)) + f(1 - p(x)\|1 - q(x))$ for some $x \sim D$. We will show that the binarised $f$-divergence is a lower bound to the categorical $f$-divergence. We have

$$
\begin{aligned}
D_f(p\|q) &= \sum_{x \sim D} f\big(p(x)\|q(x)\big) \\
&= f(p(x)\|q(x)) + \sum_{x' \sim D\setminus\{x\}} f\big(p(x')\|q(x')\big) \qquad \text{splitting the divergence into two parts}
\end{aligned}
$$

Since the first term of the split $D_f$ and $D_f^{\text{binarised}}$ is the same, it remains to show that

$$
f(1 - p(x)\|1 - q(x)) \leq \sum_{x' \sim D\setminus\{x\}} f\big(p(x')\|q(x')\big)
$$

We expand $1 - p(x)$ and $1 - q(x)$ and insert $q(x)g\left(\frac{p(x)}{q(x)}\right) = f(p(x)\|q(x))$

$$
g\left(\frac{\sum_{x' \sim D\setminus\{x\}} p(x')}{\sum_{x' \sim D\setminus\{x\}} q(x')}\right) \cdot \sum_{x' \sim D\setminus\{x\}} q(x') \leq \sum_{x' \sim D\setminus\{x\}} g\left(\frac{p(x')}{q(x')}\right) q(x')
$$

which is true by Jensen's inequality since $g$ is convex. $\qquad\square$

Alternatively, it follows from the data processing inequality that the binarised $f$-divergence is a lower bound to the categorical $f$-divergence since post-processing (by merging all but two options) can only decrease the divergence. Since $D_f^{\text{binarised}}(p\|q)$ is an $f$-divergence, the important properties that $D_f^{\text{binarised}}(p\|q) = 0$ iff $p = q$ and $D_f^{\text{binarised}}(p\|q) > 0$ otherwise also hold.

# F  A Variant of ALM Which Explicitly Takes Tokenization Bias into Account

We initially experimented with a version of ALM which explicitly selects chunks with low approximate differences in tokenization bias. This was intented to make the likelihood comparison more accurate by minimizing spurious differences caused by differing tokenization biases between the teacher and the student sequences. We did not end up choosing this formulation since it requires a pre-computation step like MinED, which results in additional complexity and prevents online applications such as Use Case 3 while performing roughly equally well. Nonetheless, we believe this previous formation and the associated formalisms may be useful to the community, and thus provide a description in the following, starting with some additional preliminaries.

**Measuring Tokenization Bias.** We can measure tokenization bias via the notion of cover encodings (Phan et al., 2024). The cover encodings of a sequence of tokens $\boldsymbol{t} = T(\boldsymbol{x})$ are all sequences of tokens $\boldsymbol{u}$ such that (i) decoding and re-encoding the sequence leads to the same token sequence $T(D(\boldsymbol{u})) = \boldsymbol{u}$ (this has been referred to as *validity*), (ii) $\boldsymbol{x}$ is a prefix of $D(\boldsymbol{u})$ and (iii) $\boldsymbol{x}$ is not a prefix of $D(\boldsymbol{u}_{0:|\boldsymbol{u}|-1})$ (i.e., the last token in $\boldsymbol{u}$ covers a part of $\boldsymbol{x}$).[18] We denote the set of all cover encodings as $\text{cover}(\boldsymbol{t})$. Note that $\boldsymbol{t} \in \text{cover}(\boldsymbol{t})$. We further define the set of *implied exclusions* $\mathcal{X}(\boldsymbol{t}) = \{D(\boldsymbol{u}) \mid \boldsymbol{u} \in \text{cover}(\boldsymbol{t}), \boldsymbol{u} \neq \boldsymbol{t}\}$. $|\mathcal{X}(\boldsymbol{t})|$ is a concrete measure of the tokenization bias of any particular token sequence $\boldsymbol{t}$. If we observe the token sequence $\boldsymbol{t} = T(\boldsymbol{x})_{:i}$, we know that $\boldsymbol{x} \notin \mathcal{X}(\boldsymbol{t})$. Otherwise, $\boldsymbol{x}$ would have been tokenized differently. In our previous example, `_Hello_World` $\in \mathcal{X}(\{$`_Hello, _Wor`$\})$.

---

[18] See Phan et al. (2024) for a more detailed exposition of cover encodings.

**Choosing Chunks with Low Approximate Tokenization Bias Difference.** We would now like to constrain chunk alignment such that the prefix and continuation are biased in the same way, i.e., $\mathcal{X}(T_S(\boldsymbol{y})) = \mathcal{X}(T_T(\boldsymbol{y}))$ and $\mathcal{X}(T_S(\boldsymbol{z})) = \mathcal{X}(T_T(\boldsymbol{z}))$. Unfortunately, this requirement is prohibitively stringent in practice: for example, if the student tokenizer is unbiased while the teacher tokenizer is not, it might rarely (if ever) be fulfilled. We thus define a relaxation which lets us define the alignment constraint $c(i, j, k, l)$ (c.f. Equation Alignment Indices) such that the chunk bias differences are *small enough*, but not necessarily zero. To this end, we introduce a precomputable scalar approximation of the difference in tokenization bias between two sequences. Let us consider arbitrary language models $A$ and $B$ (not necessarily a teacher and a student). We first define $(\mathcal{X}_A \setminus \mathcal{X}_B)(\boldsymbol{x})$ to be the set difference of implied exclusions $\mathcal{X}$ on some text $\boldsymbol{x}$ between the token sequences of the two i.e. $(\mathcal{X}_A \setminus \mathcal{X}_B)(\boldsymbol{x}) = \mathcal{X}(T_A(\boldsymbol{x})) \setminus \mathcal{X}(T_B(\boldsymbol{x}))$. We can then quantify a scalar difference $b_{A\|B}(\boldsymbol{x})$ in tokenization bias between the two sequences.

$$b_{B\|A}(\boldsymbol{x}) = \sum_{\boldsymbol{d} \in (\mathcal{X}_A \setminus \mathcal{X}_B)(\boldsymbol{x})} p_A(T_A(\boldsymbol{d}))$$

If $(\mathcal{X}_A \setminus \mathcal{X}_B)(\boldsymbol{x}) = \emptyset$, $T_A(\boldsymbol{x})$ is biased to a lesser or equal extent as $T_B(\boldsymbol{x})$ and $b_{B\|A}(\boldsymbol{x}) = 0$. If not, $b_{B\|A}(\boldsymbol{x})$ will be high if any text $\boldsymbol{d}$ which $T_A$ is biased against but $T_B$ is not is assigned a high probability by the language model $A$. Instead, if a text $\boldsymbol{d} \in (\mathcal{X}_A \setminus \mathcal{X}_B)(\boldsymbol{x})$ is assigned a low likelihood, it affects the bias difference to a lesser extent. We can use the symmetrised bias $b_{A,B}(\boldsymbol{x}) = \max(b_{B\|A}(\boldsymbol{x}), b_{A\|B}(\boldsymbol{x}))$ as a measure of the 'biasedness' of any text. However, computing $b_{B\|A}(\boldsymbol{x})$ is prohibitively expensive since we need to compute the cover encodings of every text $\boldsymbol{x}$ under both tokenization functions, as well as the likelihood assigned by the language model to every text $\boldsymbol{d}$ in the difference between implied exclusions $(\mathcal{X}_A \setminus \mathcal{X}_B)(\boldsymbol{x})$. Therefore, we make two key approximations to estimate $\hat{b}_{B\|A}(\boldsymbol{x}) \approx b_{B\|A}(\boldsymbol{x})$ (and analogously $\hat{b}_{B\|A}(\boldsymbol{x}) \approx b_{B\|A}(\boldsymbol{x})$) at nearly zero computational overhead.

**Approximating $\mathcal{X}(T(\boldsymbol{x}))$ via $\mathcal{X}(T(\boldsymbol{x})_n)$.** We approximate the implied exclusions of $T(\boldsymbol{x})$ via the implied exclusions of the last token $T(\boldsymbol{x})_n$ in $T(\boldsymbol{x})$.

$$\hat{\mathcal{X}}(T(\boldsymbol{x})) = \{D(T(\boldsymbol{x})_{<n}) \odot \boldsymbol{u} \mid \boldsymbol{u} \in \mathcal{X}(T(\boldsymbol{x})_n)\} \qquad \text{(Bias Approx. 1)}$$

This makes it possible to precompute the implied exclusions $\mathcal{X}(t)$ for all $t \in \mathcal{V}$.

**Approximating $p(T(\boldsymbol{x}))$ via $p_{\text{unigram}}(T(\boldsymbol{x})_n)$.** We approximate the language model likelihood $p(T(\boldsymbol{x}))$ via the unigram likelihood $p_{\text{unigram}}(T(\boldsymbol{x})_n)$ computed via the token counts of a text corpus.

$$\hat{p}(T(\boldsymbol{x})) = p_{\text{unigram}}(T(\boldsymbol{x})_n) \qquad \text{(Bias Approx. 2)}$$

This approximation is crude by necessity since it is distinctly intractable to run every token sequence $\boldsymbol{d} \in (\mathcal{X}_A \setminus \mathcal{X}_B)(\boldsymbol{x})$ through a large language model. However, we show later in Section 4 that it performs well in practice. Bias Approx. 1 and Bias Approx. 2 together make the bias only depend on the last token in the sequence. Going back to our distillation setup, the above approximations allow precomputing the bias difference $\hat{b}_{S,T}(\boldsymbol{x})$ for any text by enumerating all possible last token pairs $(t_S, t_T) \in \mathcal{V}_S \times \mathcal{V}_T$ and storing the result in a sparse matrix in $\mathbb{R}^{|\mathcal{V}_S| \times |\mathcal{V}_T|}$. This achieves our goal of defining an efficiently computable scalar metric to measure the difference in tokenization bias between two token sequences, and lets us set the alignment constraint to $c(i, j, k, l) := \max(\hat{b}_{S,T}(D(T_T(\boldsymbol{x})_{:i})), \hat{b}_{S,T}(D(T_T(\boldsymbol{x})_{i:j}))) \le \gamma$ where $\gamma$ is a manually defined bias difference threshold. Together, these approximations give a result in a viable alternative formulation of ALM, although one which requires pre-computation. See Table 11 for a minimal comparison of this variant with our main method.

# G Overview of Existing Cross-Tokenizer Distillation Methods

**ULD (Boizard et al., 2025)** trains the student via next-token prediction plus a Wasserstein distance loss (Kantorovich, 1960) between the teacher logits and the student logits at every step. This ensures that the absolute difference between the sorted teacher logits and sorted student logits at every position is small.

Table 11: Comparing a variant of ALM which explicitly aims to take tokenization bias into account (Appendix F) with the main ALM version on transfer of Gemma2 2B IT to the Qwen tokenizer as in Use Case 1, but training for 5k steps instead.

| ALM Setting | Benchmark | | | | | | | Avg. |
|---|---|---|---|---|---|---|---|---|
| | PiQA | ARC-C | BoolQ | MMLU | AGI-EN | AGI-ZH | IFEval | |
| ALM | 76.8 | **48.3** | **83.1** | **53.6** | 38.9 | **32.2** | **56.0** | **55.6** |
| ALM – Appendix F Variant | **76.9** | 46.8 | **83.1** | **53.6** | **39.4** | 31.5 | 52.9 | 54.9 |

Table 12: Ensembling results across Qwen and Llama pivots and across keeping/transferring the special tokens. We adjust our notation as follows: $\mathcal{X}_{\to\mathcal{Y}'}$ now denotes transferring to tokenizer $\mathcal{Y}$, while preserving the special tokens of $\mathcal{X}$ (the default in Table 2). $\mathcal{X}_{\to\mathcal{Y}}$ denotes transferring regular *and* special tokens to tokenizer $\mathcal{Y}$. We also explicitly denote the pivot used by the MinED baseline.

| | Size | ID | Model | Benchmark | | | | | | | Avg. |
|---|---|---|---|---|---|---|---|---|---|---|---|
| | | | | PiQA | ARC-C | BoolQ | MMLU | AGI-EN | AGI-ZH | IFEval | |
| Original Models | 2.6B | $\mathcal{G}$ | Gemma2 2B IT | 79.6 | 50.4 | 83.8 | 56.9 | 42.1 | 30.7 | 62.5 | 58.0 |
| | 3.2B | $\mathcal{L}$ | Llama3.2 3B IT | 76.9 | 43.9 | 78.8 | 62.4 | 36.6 | 40.2 | 76.6 | 59.3 |
| | 1.5B | $\mathcal{Q}$ | Qwen2.5 1.5B IT | 76.3 | 43.2 | 77.9 | 60.1 | 45.0 | 54.2 | 46.3 | 57.6 |
| Transfers | 2.4B | $\mathcal{G}_{\to\mathcal{Q}'}$ | $\mathcal{G}\to\mathcal{Q}$ Tok. | 76.8 | 49.0 | 82.7 | 53.6 | 38.9 | 31.6 | 53.2 | 55.1 |
| | 2.4B | $\mathcal{G}_{\to\mathcal{Q}}$ | $\mathcal{G}\to\mathcal{Q}$ Tok. + Special Tok. | 68.0 | 32.0 | 71.3 | 39.6 | 31.7 | 28.8 | 53.1 | 46.4 |
| | 3.3B | $\mathcal{L}_{\to\mathcal{Q}'}$ | $\mathcal{L}\to\mathcal{Q}$ Tok. | 77.3 | 45.6 | 79.0 | 61.6 | 37.1 | 33.3 | 76.3 | 58.6 |
| | 3.3B | $\mathcal{L}_{\to\mathcal{Q}}$ | $\mathcal{L}\to\mathcal{Q}$ Tok. + Special Tok. | 75.5 | 46.0 | 78.4 | 60.1 | 36.8 | 32.3 | 74.0 | 57.6 |
| | 2.4B | $\mathcal{G}_{\to\mathcal{L}'}$ | $\mathcal{G}\to\mathcal{L}$ Tok. | 77.1 | 48.4 | 83.0 | 53.7 | 39.2 | 32.0 | 53.8 | 55.3 |
| | 2.4B | $\mathcal{G}_{\to\mathcal{L}}$ | $\mathcal{G}\to\mathcal{L}$ Tok. + Special Tok. | 77.1 | 48.0 | 82.9 | 53.7 | 38.9 | 31.8 | 48.5 | 54.4 |
| | 1.5B | $\mathcal{Q}_{\to\mathcal{L}'}$ | $\mathcal{Q}\to\mathcal{L}$ Tok. | 76.3 | 45.9 | 78.4 | 59.6 | 44.7 | 41.9 | 45.9 | 56.1 |
| | 1.5B | $\mathcal{Q}_{\to\mathcal{L}}$ | $\mathcal{Q}\to\mathcal{L}$ Tok. + Special Tok. | 76.1 | 44.9 | 77.7 | 59.3 | 44.2 | 45.7 | 44.7 | 56.1 |
| Ensembles | 7.3B | | $\mathcal{Q}+\mathcal{G}+\mathcal{L}$ (MinED-p, $\mathcal{Q}$ pivot) | 76.9 | 46.2 | 83.2 | 64.1 | 40.6 | 39.4 | **65.9** | 59.5 |
| | 7.3B | | $\mathcal{L}+\mathcal{G}+\mathcal{Q}$ (MinED-p, $\mathcal{L}$ pivot) | 76.9 | 47.2 | 83.1 | **64.2** | 38.5 | 37.7 | 54.9 | 57.5 |
| | 7.2B | | $\mathcal{Q}+\mathcal{G}_{\to\mathcal{Q}}+\mathcal{L}_{\to\mathcal{Q}}$ (p) | 75.6 | 45.6 | 77.3 | 62.1 | 42.0 | 44.9 | 49.1 | 56.6 |
| | 7.2B | | $\mathcal{L}+\mathcal{G}_{\to\mathcal{L}}+\mathcal{Q}_{\to\mathcal{L}}$ (p) | 77.5 | **48.7** | 83.7 | 62.9 | 43.3 | 42.5 | 64.8 | 60.5 |
| | 7.2B | | $\mathcal{Q}+\mathcal{G}_{\to\mathcal{Q}'}+\mathcal{L}_{\to\mathcal{Q}'}$ (p) | 77.5 | 48.0 | 83.4 | 63.0 | **43.6** | **45.7** | 64.3 | **60.8** |
| | 7.2B | | $\mathcal{L}+\mathcal{G}_{\to\mathcal{L}'}+\mathcal{Q}_{\to\mathcal{L}'}$ (p) | **77.6** | 48.0 | **84.1** | 63.2 | **43.6** | 43.3 | 62.3 | 60.3 |
| Larger Models | 7.6B | | Llama3.1 8B IT | 81.6 | 53.9 | 85.4 | 68.4 | 47.3 | 47.1 | 81.7 | 66.5 |
| | 8.0B | | Qwen2.5 7B IT | 80.4 | 54.3 | 86.4 | 71.7 | 57.6 | 72.6 | 76.6 | 71.4 |

**MinED (Wan et al., 2024)** aligns the student and teacher logits by greedily aligning every token in the students' vocabulary with the teacher token which has the minimal Levenshtein distance (Levenshtein, 1966) to the student token. The main objective is again minimizing cross-entropy of the next-token prediction. The KL-divergence between the student logits and aligned teacher logits is added at every position where there is a one-to-one alignment between the elements of the two sequences.

**DSKD (Zhang et al., 2024b)** projects the student representations to the teacher sequence and the teacher representations to the student sequence via a cross-attention mechanism, then adds the KL-divergence between the teacher predictions and the student-to-teacher projected predictions (and vice versa) to the next-token prediction loss.

# H    Analysing the Choice of Ensemble Pivot & Special Tokens

For our main ensembling results (Table 2), we have chosen to transfer to the Qwen tokenizer, i.e., to use Qwen as the 'pivot'. The question arises, then, of how the choice of pivot impacts performance. We additionally transfer Gemma2 2B and Qwen2.5 1.5B to the Llama3 tokenizer and ensemble with the (non-transferred) Llama3 3B to answer this question. Results are shown in Table 12. We find that, at an average score of 60.8 with Qwen as the pivot and 60.3 with Llama as the pivot, the ensemble of transferred models is robust to the choice of pivot. This is, however, not true for the MinED baseline: here, using Llama as the pivot catastrophically degrades the ensemble to perform on average worse than the individual ensemble constituents.

We use this opportunity to additionally study a related subtle choice: whether to keep the special tokens (such as <bos>, <eos>, etc.) of the original tokenizer, or to use the special tokens of the new

tokenizer upon transfer. Different special tokens do not pose a problem for sequence alignment since they can be easily detected and manually aligned. The same holds true for autoregressive generation, where it is fairly trivial to maintain separate running token sequences for the different models, which each insert their own special tokens as needed when one is generated. Nonetheless, this does result in implementation overhead, and for some use cases it may be convenient to achieve 100% parity with the target tokenizer by transferring the special tokens as well. Alongside the aforementioned ensembling results, Table 12 also shows the performance of models which had their special tokens transferred and ensembles of these models. For the Llama pivot, transferring the special tokens completely preserves performance. However, for the Qwen pivot, transferring the special tokens catastrophically degrades performance (especially the performance of the Gemma2 2B constituent). This is the case since Qwen2 does not use a beginning-of-sequence <bos> token. In models which do use a <bos> token, this token is often heavily attended to, behaving as an 'attention sink' (Barbero et al., 2025). The models transferred to the Qwen2 special tokens thus have to relearn to distribute attention from the (now absent) <bos> to different tokens in the sequence. Evidently, Llama3 is able to cope with this situation better than Gemma2; more training of Gemma2 may fix this problem. Based on these findings, we preserve the special tokens in our main experiments since it is in general the safer choice, and the only downside for our purpose is additional implementation complexity.

# I   Implementing ALM

ALM is simple to implement efficiently by making changes to the data preparation and the loss computation.

**Data preparation.**   Alongside tokenizing all example texts in the current batch, data preparation must include computing aligned chunks of tokens between the teacher and the student. To implement the chunk alignment algorithm, we can take advantage of the fact that both token sequences encode the same bytes. It is thus possible to compute alignments in a single pass via indices $(i, k)$, where $i$ is increased as long as the token at position $i$ in one sequence ends earlier than the token at position $j$ in the other sequence, and vice versa, and indices are stored if both tokens end at the same byte position. This is in contrast to more complex approaches used in prior work (Fu et al., 2023; Wan et al., 2024): they minimize edit distance between alignments via variants of the Needleman-Wunsch algorithm (Needleman & Wunsch, 1970). This is not necessary if the tokenizers have first been converted to the byte level. We can represent the computed chunks via binary matrices $M \in \{0, 1\}^{b \times m \times k}$ and $N \in \{0, 1\}^{b \times n \times k}$ where $b$, $m$, $n$ are the batch size, the student sequence length and the teacher sequence length, respectively and $k = \min(m, n)$ is the maximum amount of chunks (i.e. the minimum of the student and teacher sequence lengths). $\mathbf{M}$ and $\mathbf{N}$ have entries one if the i-th token is part of the j-th chunk and zero otherwise. The above can be done in a separate data-loading thread to avoid blocking the main process.

**Loss computation.**   Loss computation involves computing the student and teacher next-token log probabilities over input tokens $\log p_S(\boldsymbol{y}|\boldsymbol{x}) \in \mathbb{R}^{b \times m}$ and $\log p_T(\boldsymbol{y}|\boldsymbol{x}) \in \mathbb{R}^{b \times n}$ where $\boldsymbol{y}$ are the input tokens shifted by one as in the causal language modelling objective. We can then compute chunk log-probabilities $\boldsymbol{C}_S \coloneqq (\log p_S(\boldsymbol{y}|\boldsymbol{x}))\boldsymbol{M}$ and $\boldsymbol{C}_T \coloneqq (\log p_T(\boldsymbol{y}|\boldsymbol{x}))\boldsymbol{N}$. We then apply the chunk-level $f$-divergence element-wise to $\boldsymbol{C}_S$ and $\boldsymbol{C}_T$ and take the mean to compute the loss. In the simplest case where $f = f_{\text{TVD}}$ and $\tau \to \infty$, this reduces to $\mathcal{L}_{S,T}^{\text{ALM}}(\boldsymbol{x}) = \frac{1}{b \cdot k} \sum_{i,j} |(\boldsymbol{C}_S)_{\text{i,j}} - (\boldsymbol{C}_T)_{\text{i,j}}|$. All of the above results in negligible overhead compared to the forward and backward pass over the Transformer backbone (two gathers, two matrix multiplications, and some element-wise comparisons). Alignment constraints $c(i, j, k, l)$ (c.f. Equation Alignment Indices) and chunk debiasing (c.f. Equation Outcome-Debiased Chunk-Level Probability) add some implementation complexity to the above but are cheap to compute and, in principle, optional.

