# OpenReview forum: "Universal Cross-Tokenizer Distillation via Approximate Likelihood Matching"
_NeurIPS.cc/2025/Conference — NeurIPS 2025 poster_

### Official Review · Reviewer_bnAG · 2025-07-01

**Clarity:** 3
**Significance:** 4
**Originality:** 3
**Rating:** 5
**Confidence:** 4

**Summary:**

This paper focuses on a scenario where tokenizers for teacher and student LLMs are different in distillation. To address this challenging scenario, this paper proposes a new approach called approximate likelihood matching (ALM). ALM primarily tries to minimise the divergence between teacher and student at a chunk level, given a sequence. This effectively allows the student's likelihood to be close to the teacher's likelihood, and this chunk-based approach apparently enables cross-tokenizer distillation, which has not been solved in the previous literature. The experiments are divided into three parts: (i) a self-distillation scenario; (ii) large-to-small domain-specialised model distillation; and (iii) training-free tokenizer adaptation. The results show the effectiveness of the proposed ALM approach for each scenario.

**Questions:**

**Suggested changes and questions**
1. I think the description of chunk debiasing and a hidden state alignment objective is quite dense. I would suggest inserting a conceptual figure for better understanding.
2. On Weakness 1, I personally would like to see the results of self-distillation and ensembling experiments on a larger scale (> 10B) if resources are available. This is quite useful in that it can help measure the robustness of the proposed approach against models with far better task performances. I sometimes see approaches, which successfully minimize performance degradation in a small-scale experiment but see a larger degradation in large-scale (potentially because larger models might be more sensitive to model modifications.)
3. On Weakness 2, it would be nice to have results on non-math-related tasks.
4. It seems to me that while ALM and ZeTT are from different paradigms, both can indeed replace the tokenizer of a source model. Could the paper include a discussion on this point? For instance, I assume that training a hypernetwork is costly, so using self-distillation with ALM as in the first use case is a better option in terms of both computational costs and task performance. Is this truly the case?

**Ethical Concerns:**

["NO or VERY MINOR ethics concerns only"]

**Final Justification:**

* The authors address the first weakness with sufficient evidence (as long as the final version incorporates this discussion).
* While the second weakness has not been addressed, this point is a minor weakness, which should not affect the overall rating much.
* The additional experiment on the model with 10B or larger even seems fascinating and exciting.

**Limitations:**

yes

**Quality:**

3

**Strengths And Weaknesses:**

**Strengths**
1. This paper proposes a novel approach to allow distillation between different models with different tokenizers using chunk-based approximate log-likelihood matching.
2. The experiments are quite extensive and showcase the effectiveness of the proposed approach in three ways: self-distillation/ensembling, strong-to-weak domain-specialised model distillation, and zero-shot tokenizer transfer using a hypernetwork. This cross-tokenizer distillation must be quite a challenging scenario, and thus, observing a good performance across different scenarios suggests further potential of the proposed approach.

**Weaknesses**
1. While I acknowledge the effectiveness of the proposed approach, I think the first use case, except for ensembling, is a bit of a waste of space. Specifically, I do not see any practical value of distilling a subword-level tokenizer to a byte-level tokenizer in the first use case, as the latter neither offers a task performance advantage nor better tokenization efficiency. If this is something important, the paper warrants further clarification on this experiment.
2. While the large-to-small domain-specialised model distillation in the second use case is fascinating, the paper omits reporting general performance evaluation (e.g. evaluation on MMLU, IFEval, ARC, BoolQ, etc.). This evaluation must be essential to provide a more holistic assessment of the proposed method.

---

> ### Author Rebuttal · Authors · 2025-07-30
>
> Thanks a lot for your review! We answer your questions below.
>
> > I do not see any practical value of distilling a subword-level tokenizer to a byte-level tokenizer in the first use case, as the latter neither offers a task performance advantage nor better tokenization efficiency. If this is something important, the paper warrants further clarification on this experiment.
>
> The experiments on transfer to bytes are important for two reasons: (1) they allow us to show that ALM works well in the extreme case of effectively zero vocabulary overlap between the teacher and the student, and does thus not implicitly rely on large vocabulary overlap / similar source and target tokenizers. Even more importantly, (2) recent work such as BLT [[1]] and H-Net [[2]] pushes byte-level language modelling toward new levels of efficiency, demonstrating that they can achieve better scaling behavior than subword tokenization while avoiding the drawbacks of subword tokenization (such as the vocabulary bottleneck [[3]], tokenization bias [[4]] and imperfect character-level understanding [[5]]). By introducing an effective strategy for subword-to-byte distillation, our work could facilitate the development of byte-level models from pre-existing LLMs and speed up the experimentation with these new architectures and their adoption. We will clarify both these points in the paper.
>
> > While the large-to-small domain-specialised model distillation in the second use case is fascinating, the paper omits reporting general performance evaluation (e.g. evaluation on MMLU, IFEval, ARC, BoolQ, etc.). It would be nice to have results on non-math-related tasks.
>
> Since our goal for the experiments in Use Case 2 was specifically to create a math-specialized model, we believe the performance on math tasks is by far the most important. However, we do agree that evaluation on a larger set of benchmarks would be beneficial, and we will add this to the camera-ready (our resources were occupied by transferring a larger model via ALM so we were not able to run this during the rebuttal period).
>
> > I personally would like to see the results of self-distillation and ensembling experiments on a larger scale (> 10B) if resources are available.
>
> We agree on the importance of scaling up, and we ran self-distillation experiments on transferring Gemma3 12B to the Qwen3 tokenizer to address this point. The results are shown below.
>
> |                                  | PiQA | ARC-C | BoolQ | MMLU | AGI-EN | AGI-ZH | IFEval | Avg. |
> |----------------------------------|------|-------|-------|------|--------|--------|--------|------|
> | Gemma3-12B-Instruct                    | 78.2 | 60.1  | 87.5  | 71.5 | 60.0   | 57.4   | 83.3   | 71.2 |
> | $\rightarrow$ Qwen3 Tok. via SFT | 81.4 | 56.7  | 83.7  | 67.2 | 46.2   | 43.5   | 73.4   | 64.6 |
> | $\rightarrow$ Qwen3 Tok. via ALM | 78.8 | 59.9  | 86.9  | 70.7 | 57.6   | 53.6   | 76.7   | 69.2 |
>
> The hyperparameters exactly match those of the subword $\rightarrow$ subword tokenizer transfer experiments in the paper, and the loss continuously decreases throughout training (no loss spikes) for both methods. Notably, while performance on some benchmarks increases throughout training for the SFT baseline (ARC-C, PiQA, IFEval), performance on others stagnates or even decreases (BoolQ, MMLU, AGI-*). From these results, we can conclude that ALM is superior to SFT for model scales larger than those originally considered in our paper.
>
> > I think the description of chunk debiasing and a hidden state alignment objective is quite dense. I would suggest inserting a conceptual figure for better understanding.
>
> Thank you for raising this point. We agree that a figure should ease understanding and will add one to the final version of the paper.
>
> > It seems to me that while ALM and ZeTT are from different paradigms, both can indeed replace the tokenizer of a source model. Could the paper include a discussion on this point? For instance, I assume that training a hypernetwork is costly, so using self-distillation with ALM as in the first use case is a better option in terms of both computational costs and task performance. Is this truly the case?
>
> This is another excellent point. It is correct that if the goal is transferring to a single target tokenizer, direct tokenizer transfer self-distillation via ALM should be the most efficient. However, if the goal is to potentially transfer to up to N different tokenizers, there is some threshold of N where we expect it would become more efficient to train a ZeTT hypernetwork once, then have a brief additional training phase per tokenizer, instead of separately transferring to the N tokenizers individually via ALM self-distillation. We will add this intuition to the section on Use Case 3 in the paper.
>
> Please let us know if you have any additional questions!
>
> [1]: https://arxiv.org/abs/2412.09871
> [2]: https://arxiv.org/abs/2507.07955
> [3]: https://arxiv.org/abs/2301.10472
> [4]: https://arxiv.org/abs/2506.14123
> [5]: https://arxiv.org/abs/2409.15452

---

> > ### Comment · Reviewer_bnAG · 2025-08-01
> > **Thanks for your response**
> >
> > Many thanks for your point-to-point and concise response. All the responses have addressed my concerns and questions effectively. I have no additional questions and look forward to seeing the final version reflecting our discussion.

---

### Official Review · Reviewer_sqsY · 2025-07-03

**Clarity:** 3
**Significance:** 2
**Originality:** 4
**Rating:** 4
**Confidence:** 4

**Summary:**

This paper introduces Approximate Likelihood Matching (ALM), a novel cross-tokenizer distillation method designed to transfer knowledge between large language models (LLMs) with fundamentally different tokenizers. ALM enables effective distillation across diverse tokenizer types, such as from subword to byte-level tokenizers, by minimizing the differences in chunk likelihoods between the teacher and student models. The method incorporates optional debiasing techniques to mitigate tokenization bias and aligns hidden states to enrich the teacher signal. The authors validate ALM's efficacy through three distinct use cases: tokenizer transfer via self-distillation, large-to-small model distillation, and zero-shot tokenizer transfer hypernetwork training. The results demonstrate ALM's ability to outperform prior methods, significantly expanding the range of teacher-student pairs for distillation and enhancing the adaptability and transferability of LLMs.

**Questions:**

The questions I expect to ask would be similar to the above section.

**Ethical Concerns:**

["NO or VERY MINOR ethics concerns only"]

**Final Justification:**

Considering the effectiveness and novelty of the method proposed in this article and the opinions of other reviewers, I give the final rating of borderline acceptance.

**Limitations:**

The authors have discussed some limitations of their work, such as the computational constraints that limited their experiments to models ranging from 2.4B to 8B parameters and a relatively small number of training tokens (≈0.6B). They also noted that their method has only been tested in the supervised KD setup and its application to more intricate distillation techniques remains future work. However, there are a few additional points that could be further addressed:
1. Scalability and Larger Models: The experiments were limited to models with up to 8B parameters. The paper does not provide sufficient analysis on how ALM would perform with larger models or more extensive training data, which limits the understanding of its generalizability to more complex and larger-scale applications.
2. Robustness to Tokenization Bias: While ALM handles tokenizer mismatch effectively, its resilience to residual tokenization bias—especially in low-resource or non-Latin languages—remains underexplored. Evaluating performance under more diverse tokenization schemes would strengthen the generality of the method.

**Paper Formatting Concerns:**

I have no issues.

**Quality:**

3

**Strengths And Weaknesses:**

Strengths:
1. Innovative Distillation Method: The paper introduces ALM, a novel cross-tokenizer distillation method that enables effective knowledge transfer between LLMs with different tokenizers.
2. Empirical Validation: The authors provide comprehensive experiments across three distinct use cases, demonstrating ALM's effectiveness and superiority over existing methods.
3. Technical Clarity: The methodology is clearly explained, making it accessible and reproducible for other researchers.

Weaknesses:
1. Limited Scalability: The experiments are restricted to models with up to 8B parameters, limiting the understanding of ALM's performance on larger models.
2. Comparative Analysis: The paper lacks detailed comparisons with other state-of-the-art methods focusing on similar domain sampling, which could better illustrate ALM's novelty and effectiveness.
3. Broader Contextualization: The paper does not fully contextualize ALM within the broader landscape of language model distillation, including its relation to other emerging trends and challenges.

---

> ### Author Rebuttal · Authors · 2025-07-30
>
> Thank you for your review! We address your concerns below.
>
> > The experiments are restricted to models with up to 8B parameters, limiting the understanding of ALM's performance on larger models.
>
> We favored breadth for our experiments (thorough ablations, sensitivity analysis, etc.) over scaling to large parameter counts. However, we do recognize the importance of showing that our method scales to larger LLMs. To do so, **we have conducted additional experiments transferring a larger LLM, Gemma3-12B, to the Qwen3 tokenizer**. The results are shown below.
>
> |                                  | PiQA | ARC-C | BoolQ | MMLU | AGI-EN | AGI-ZH | IFEval | Avg. |
> |----------------------------------|------|-------|-------|------|--------|--------|--------|------|
> | Gemma3-12B-Instruct                    | 78.2 | 60.1  | 87.5  | 71.5 | 60.0   | 57.4   | 83.3   | 71.2 |
> | $\rightarrow$ Qwen3 Tok. via SFT | 81.4 | 56.7  | 83.7  | 67.2 | 46.2   | 43.5   | 73.4   | 64.6 |
> | $\rightarrow$ Qwen3 Tok. via ALM | 78.8 | 59.9  | 86.9  | 70.7 | 57.6   | 53.6   | 76.7   | 69.2 |
>
> The hyperparameters exactly match those of the subword $\rightarrow$ subword tokenizer transfer experiments in the paper, and the loss continuously decreases throughout training (no loss spikes) for both methods. Notably, while performance on some benchmarks increases throughout training for the SFT baseline (ARC-C, PiQA, IFEval), performance on others stagnates or even decreases (BoolQ, MMLU, AGI-*). From these results, we can conclude that ALM is superior to SFT for model scales larger than those originally considered in our paper.
>
> > The paper lacks detailed comparisons with other state-of-the-art methods focusing on similar domain sampling, which could better illustrate ALM's novelty and effectiveness.
>
> Could you give an example of one of the ‘similar domain sampling’ methods you are referring to? Sampling the distillation training data in a way that closely resembles the LLM pretraining data is generally hard since the LLM training data mix is unknown. Furthermore, **ALM is orthogonal to the way the data is sampled; we would expect improvements to the data sampling technique to synergize with the improvements from ALM**.
>
> > The paper does not fully contextualize ALM within the broader landscape of language model distillation, including its relation to other emerging trends and challenges.
>
> **We contextualize ALM within the broader landscape of distillation in Section 2.3, and provide a detailed comparison with prior cross-tokenizer distillation methods in Appendix G**. An exhaustive overview over the ‘emerging trends and challenges’ in language model distillation (besides cross-tokenizer distillation) would have gone outside the scope of our paper. We will add a reference to the survey by Xu et al. [[1]] to Section 2.3 as an additional source of information about the broader language model distillation landscape.
>
> > Robustness to Tokenization Bias: While ALM handles tokenizer mismatch effectively, its resilience to residual tokenization bias—especially in low-resource or non-Latin languages—remains underexplored. Evaluating performance under more diverse tokenization schemes would strengthen the generality of the method.
>
> **Our benchmark results on Chinese (e.g. in Table 2) indicate that ALM can effectively preserve performance in non-Latin scripts (and even in languages without clear word boundaries)**. We believe these results sufficiently support that ALM is not restricted to the Latin script.
>
> We hope this addresses your concerns, and please let us know if you have any further questions.
>
> [1]: https://arxiv.org/abs/2402.13116

---

> > ### Comment · Reviewer_sqsY · 2025-08-06
> >
> > Thank you for the author's detailed response, which has addressed several of my concerns. I will therefore maintain my final rating.

---

### Official Review · Reviewer_G5Di · 2025-07-05

**Clarity:** 3
**Significance:** 3
**Originality:** 3
**Rating:** 4
**Confidence:** 2

**Summary:**

The paper tackles the problem of current distillation methods that require similar tokenizers between teacher and student LLMs. The authors propose a novel cross-tokenizer distillation method ALM, enabling effective knowledge transfer even when the teacher and student use different tokenizers. The approach is evaluated on three distinct use cases, consistently outperforming existing methods.

**Questions:**

1. Does the teacher and student models have to have similar tokenizers? What if the teacher and student tokenizers have very small overlappings?
2. Is it possible that for some tokens, they are not covered by chunks from the teacher side, and therefore students could not learn anything on them?
3. What if the text is very long and the greedy alignments become sub optimal?

**Ethical Concerns:**

["NO or VERY MINOR ethics concerns only"]

**Limitations:**

The computational complexity of the proposed method could be pretty high. The greedy alignments could be sub optimal when text is long.

**Quality:**

3

**Strengths And Weaknesses:**

Strengths:
1. The problem studied in the paper is important and critical. The differences between tokenizers are significant blocker between knowledge transferring between models. Having a method to overcome such limitations could directly benefit many applications, such as distilling general, large models for domain-specific use smaller models.
2. The proposed solution ALM, which aims to model the token level matching probability via chunk likelihoods, makes sense to me. It is a sound methodology.
3. The paper is easy to follow, especially for readers who have never touched the problem of cross-tokenizer distillation. The authors give sufficient information in the preliminaries and background section.

Weaknesses
1. From the paper, the complexity of computing loss function on chunk-level probabilities is pretty high. So I am not sure if it is practical to actually deploy the proposed ALM into production use cases.
2. The first eq in section 3 (Alignment Indices) is not very clear. The authors use a greedy approach to get token chunks. However, when the text is long, there can be multiple valid ways to segment and align text between two different tokenizers. Therefore, greedy strategy might not find the optimal segmentation.
3. The paper also didn't mention how the student could learn on tokens not covered by chunks from teacher model.

---

> ### Author Rebuttal · Authors · 2025-07-30
>
> Thank you for very much for your review! We believe your concerns mostly stem from a couple of misunderstandings which we hope to clarify here (and will also clarify in the paper).
>
> > From the paper, the complexity of computing loss function on chunk-level probabilities is pretty high. So I am not sure if it is practical to actually deploy the proposed ALM into production use cases.
>
> Complexity (computational and implementation-wise) indeed plays an important role. **Chunk alignments can be computed in O(n) via a simple dynamic programming algorithm** and represented as an $b \times m \times k$ and $b \times n \times k$ matrix (where $m$ and $n$ are the teacher and student sequence lengths, respectively, $k$ is the maximum amount of chunks, i.e. $\min(m,n)$ and $b$ is the batch size) with entries 1 if the i-th token is part of the j-th chunk, and zero otherwise (an example is shown below).
>
> Example Matrix 1:
>
> | The | distill | ation | procedure |
> |-----|---------|-------|-----------|
> | 1   |         |       |           |
> |     | 1       | 1     |           |
> |     |         |       | 1         |
>
> Example Matrix 2:
>
> | The | dist | illation | proc | edure |
> |-----|------|----------|------|-------|
> | 1   |      |          |      |       |
> |     | 1    | 1        |      |       |
> |     |      |          | 1    | 1     |
>
> These matrices can be precomputed in the dataloader. Afterwards, ALM is easy to implement:
>
> Compute the teacher and student log-probabilities ($b \times m \times |V_1|$ and $b \times n \times |V_2|$).
> Select the teacher and student log-probabilities along the true next tokens in the data ($b \times m$ and $b \times n$, respectively).
> Matrix-multiply the log-probabilities with Matrix 1 and Matrix 2 to compute chunk-level log-probabilities ($b \times k$ and $b \times k$).
> Compute the loss as the mean of some distance function between the chunk-level probabilities (for example, the mean of the absolute differences between the chunk-level probabilities).
>
> **All of this is implementable in a couple of lines of code in PyTorch (and will be open-sourced along with the rest of our code)**. We will add an Appendix to the paper to discuss these implementation considerations.
>
> > The first eq in section 3 (Alignment Indices) is not very clear. The authors use a greedy approach to get token chunks. However, when the text is long, there can be multiple valid ways to segment and align text between two different tokenizers. Therefore, greedy strategy might not find the optimal segmentation.
>
> There are indeed many valid ways to align text between two different tokenizers (exponentially many in the sequence length). However, **there is only one greedy segmentation, the one with the most (and thus shortest possible) chunks**. Our intuition behind selecting this segmentation is that it provides the most signal: for any non-greedy segmentation, we could break down a chunk into two or more chunks which then convey more information about the teacher's predictions of the tokens in this chunk (e.g. if “The distillation” was a single chunk in the example above). It is however a valid empirical question whether non-greedy and/or partially overlapping segmentations could improve ALM; this could be an area for future work. We will clarify this in the paper.
>
> > The paper also didn't mention how the student could learn on tokens not covered by chunks from teacher model.
>
> **The student does not learn from next-token predictions which are not part of any chunk** (i.e. not along the ‘main path’, the true next token logit in the data). As discussed in Section 3.2, this does decrease the density of the signal, which however in practice does not have a large effect on our method's effectiveness and can be alleviated by adding Hidden State Alignment if desired (see Table 7).
>
> > Does the teacher and student models have to have similar tokenizers? What if the teacher and student tokenizers have very small overlappings?
>
> The Qwen and Llama tokenizers have substantial overlap (intersection over union of tokens in the vocabulary = 64%), the Gemma and Qwen tokenizers have lower overlap (intersection over union of tokens in the vocabulary = 29%). In the extreme case of transfer to bytes (Use Case 1), the overlap is ~0% since the student tokenizer only contains the 256 tokens corresponding to the possible values of a byte. **ALM can effectively cope with this case of essentially zero overlap** (see Table 1).
>
> > Is it possible that for some tokens, they are not covered by chunks from the teacher side, and therefore students could not learn anything on them?
>
> Every token in both the student’s and teacher’s token sequence is part of a chunk, so the student will learn from all tokens which appear in the input text and it is not possible for chunks not to be covered from the teacher's side (assuming both tokenizers can segment arbitrary text, which is true for virtually all modern tokenizers and can otherwise be made true; see Footnote 1).
>
> > What if the text is very long and the greedy alignments become sub optimal?
>
> As discussed above, **greedy alignments will not become suboptimal for long texts**. Greedy alignments are always the alignments which result in the chunks with the highest granularity / the lowest average token count per chunk and thus the maximal ability to transmit signal from the teachers’ predictions.
>
> We would kindly ask you to update your score if your concerns have been addressed, and please let us know if you have any further questions.

---

### Official Review · Reviewer_RibG · 2025-07-21

**Clarity:** 4
**Significance:** 3
**Originality:** 4
**Rating:** 4
**Confidence:** 3

**Summary:**

The authors introduce Approximate Likelihood Matching (ALM), a new objective for cross-tokenizer knowledge distillation. The authors discuss the three use cases to show the effectiveness of the proposed method.

**Questions:**

1 How does ALM behave when the teacher’s vocabulary is much larger than the student’s?
2 From the ablation study, it looks like the performance is very sensitive to some setups, like the learning rate and the selection heuristic. How do we determine the best setup when we are performing the proposed ALM?
4 Authors claimed that the proposed method enables effective distillation across fundamentally different tokenizers. How do we measure the distance between tokenizers? It would be good to see the trend of using the proposed method on tokenizers with increasing distances.

**Ethical Concerns:**

["NO or VERY MINOR ethics concerns only"]

**Final Justification:**

4: Borderline accept

**Limitations:**

Yes

**Paper Formatting Concerns:**

No formatting concerns.

**Quality:**

3

**Strengths And Weaknesses:**

Strengths
1 The problem definition is clear and the proposed method should be of interests for researchers in this area.
2 The authors detailed discuss the three use cases to show the effectiveness of the proposed method.

Weaknesses
1 All experiments are ≤ 8 B params and ~0.6 B tokens. The effectiveness for frontier-scale LLM is unclear.
2 Considering the small models and tokenizers, and the sensitiveness to the hyper-parameters, the performance improvement is slight.

---

> ### Author Rebuttal · Authors · 2025-07-30
>
> Thank you for your review! We address your concerns below.
>
> > All experiments are ≤ 8 B params and ~0.6 B tokens. The effectiveness for frontier-scale LLM is unclear.
>
> We favored breadth for our experiments (thorough ablations, sensitivity analysis, etc.) over scaling to large parameter counts. However, we do recognize the importance of showing that our method scales to larger LLMs. To do so, **we have conducted additional experiments transferring a larger LLM, Gemma3-12B, to the Qwen3 tokenizer**. The results are shown below.
>
> |                                  | PiQA | ARC-C | BoolQ | MMLU | AGI-EN | AGI-ZH | IFEval | Avg. |
> |----------------------------------|------|-------|-------|------|--------|--------|--------|------|
> | Gemma3-12B-Instruct                    | 78.2 | 60.1  | 87.5  | 71.5 | 60.0   | 57.4   | 83.3   | 71.2 |
> | $\rightarrow$ Qwen3 Tok. via SFT | 81.4 | 56.7  | 83.7  | 67.2 | 46.2   | 43.5   | 73.4   | 64.6 |
> | $\rightarrow$ Qwen3 Tok. via ALM | 78.8 | 59.9  | 86.9  | 70.7 | 57.6   | 53.6   | 76.7   | 69.2 |
>
> The hyperparameters exactly match those of the subword $\rightarrow$ subword tokenizer transfer experiments in the paper, and the loss continuously decreases throughout training (no loss spikes) for both methods. Notably, while performance on some benchmarks increases throughout training for the SFT baseline (ARC-C, PiQA, IFEval), performance on others stagnates or even decreases (BoolQ, MMLU, AGI-*). From these results, we can conclude that ALM is superior to SFT for model scales larger than those originally considered in our paper.
>
> > Considering the small models and tokenizers, and the sensitiveness to the hyper-parameters, the performance improvement is slight.
>
> We beg to differ as, in our view, our experiments show the opposite: across all hyperparameter settings of ALM for the learning rate, temperature, and distance function, our method outperforms the baselines across all baseline hyperparameters (see Tables 5 and 6).
> **The performance improvements close the gap toward the teacher by an additional 34% over the baselines in Use Case 1 (Figure 2), improving by 3% absolute percentage points on GSM8K in Use Case 2 (Table 3), and improving over SFT by up to 6.5% absolute percentage points in Use Case 3 (Table 4)**; we consider these margins to be quite significant.
>
> And to answer your questions.
>
> > How does ALM behave when the teacher’s vocabulary is much larger than the student’s?
>
> This is an interesting question! The extreme case of the teacher’s vocabulary being much larger than the student’s occurs in our experiments on transfer to bytes, where the student’s vocabulary has a size of just 256 tokens, whereas the teacher’s vocabulary has ~100k-200k tokens. We found that, **especially in such cases, ALM performs well while prior methods break down** (see Table 1 and the discussion of Use Case 1).
>
> > From the ablation study, it looks like the performance is very sensitive to some setups, like the learning rate and the selection heuristic.
> > How do we determine the best setup when we are performing the proposed ALM?
>
> **We use a single setting for ALM that works well across all experiments** (temperature=100, distance function=KL, threshold=0.1, LR=1e-5) as detailed in Appendix B. The only exception is transfer to bytes, where we make a couple of adjustments detailed in Appendix C. So, when using ALM, we recommend that practitioners use the hyperparameters we use across all our experiments. The ablations & sensitivity analyses are only intended to provide intuition as to the effect of varying these hyperparameters.
>
> > Authors claimed that the proposed method enables effective distillation across fundamentally different tokenizers. How do we measure the distance between tokenizers? It would be good to see the trend of using the proposed method on tokenizers with increasing distances.
>
> How to measure the distance between tokenizers is an interesting question and one which does not have a trivial answer. A simple proxy is the vocabulary size, but it might not always be accurate (and would easily break in the setting of duplicating the tokenizer and adding N unused/barely used tokens). We will clarify this in the paper and would hope to see future work on quantifying tokenizer distance.
> **Our claim to enable distillation between fundamentally different tokenizers is based on the fact that ALM allows for distilling between subword tokenizers and byte tokenization which are the two extremes of the spectrum contemporary tokenizers lie on**.  We agree that quantifying the trend of cross-tokenizer distillation method performance across increasingly distant tokenizers would be a very interesting experiment for follow-up work, thanks for the suggestion. **The reason we did not explore this setting is that we constrained our experiments to subword tokenizers which are used in practice by modern LLMs** (these are largely in the ~100k-200k vocabulary range).
>
> We hope this addresses your concerns, and please let us know if you have any further questions.

---

> > ### Comment · Reviewer_RibG · 2025-08-06
> >
> > Thanks for your response. I will increase the quality rating, but retain the overall rating
> >
> > 1 The additional experiments on Gemma3-12B are good. But it's also a relatively small model with a small tokenizer. The effectiveness of the proposed method for frontier-scale LLM is still unclear.
> > 2 I agree that the distance measurement does not have a trivial answer. Using vocabulary size is a practical proxy. Does the proposed ALM converge or perform more robustly when the student uses a very small vocabulary? Conversely, as the vocabulary grows larger, does that impact distillation quality? A qualitative trend would help understand how ALM performs in different situations.

---

> > > ### Author Response · Authors · 2025-08-07
> > >
> > > > The additional experiments on Gemma3-12B are good. But it's also a relatively small model with a small tokenizer.
> > >
> > > Although we do not agree, we suppose it is not productive to argue whether 12 billion parameters is still 'small' or not. However, we want to clarify that __Gemma3, at 256k tokens in the vocabulary, actually has a larger tokenizer than almost all other models. This means transfer to the Qwen3 tokenizer shrinks the vocabulary size by ~100k, yet ALM still quickly almost completely recovers the performance.__ Thus, although there are still always larger models to scale to, we believe our additional experiments provides strong evidence that ALM scales favorably to models substantially beyond the model sizes we considered in the paper, and which are on the same order of magnitude as the most competitive Open Source models.
> > >
> > > > I agree that the distance measurement does not have a trivial answer. Using vocabulary size is a practical proxy. Does the proposed ALM converge or perform more robustly when the student uses a very small vocabulary? Conversely, as the vocabulary grows larger, does that impact distillation quality? A qualitative trend would help understand how ALM performs in different situations.
> > >
> > > We have not conducted detailed experiments in this direction, since as per our original response, we constrained our experiments to subword tokenizers which are used in practice by modern LLMs; these are mostly around the ~100k-200k vocabulary range.
> > >
> > > We would expect distillation quality to not necessarily be correlated with the absolute vocabulary size, but potentially with the *vocabulary size difference* between the teacher and the student. Although ALM still performs well in settings with extremely high vocabulary size differences (e.g., subword $\rightarrow$ byte transfer), it is a more challenging setting. We will add this intuition to the paper, and thank you for raising this point.

---

### Note · Authors · 2025-08-12

We thank all reviewers again for their reviews. To summarize the reviewers' concerns and how we addressed them:

- Reviewers RibG, sqsY were concerned about experiments being limited to models up to 8B parameters, and reviewer bnAG noted that they would like to see experiments at a larger scale if possible. __To address this concern, we ran experiments on transfer of Gemma3-12B to the Qwen3 tokenizer, concluding that ALM can scale well beyond the model sizes we considered in the paper, and obtains a ~5% absolute gap to SFT in this case__ (see the rebuttal to reviewer RibG, sqsY, or bnAG).
- Reviewer RibG was concerned about sensitiveness to the hyperparameters. We clarified that __we use a single set of hyperparameters for all experiments (except transfer to bytes), and although the hyperparameters (such as the temperature) do impact performance, our method outperforms prior methods across almost all hyperparameter choices (see Appendix A)__.
- Reviewer G5Di had some concerns about the methodology (e.g., whether greedy alignments become suboptimal for long sequences, and about high complexity of the method). __We clarified that greedy alignments do not become suboptimal for long sequences, and that implementational as well as algorithmic complexity of our method is fairly light__. Since Reviewer G5Di did not respond to our rebuttal, we do not know whether these misunderstandings have been resolved.
- Reviewers bnAG and sqsY requested clarifications pertaining to an experiment and the broader context of the paper, respectively, which they have stated have been addressed by our rebuttal.

On the other hand, the reviewers noted that '[t]he problem definition is clear' (RibG), 'the problem studied in the paper is important and critical', 'the paper is easy to follow, especially for readers who have never touched the problem of cross-tokenizer distillation' (G5Di), 'the authors provide comprehensive experiments across three distinct use cases, demonstrating ALM's effectiveness and superiority over existing methods' (sqsY) and 'cross-tokenizer distillation must be quite a challenging scenario, and thus, observing a good performance across different scenarios suggests further potential of the proposed approach' (bnAG).

---

### Decision · Program_Chairs · 2025-09-17

**Decision:**

Accept (poster)

**Comment:**

This work develop a principled cross-tokenizer distillation method to solve this crucial deficiency in current distillation method, which typical requires similar tokenizer. All reviewers have positive ratings to this paper, and acknowledge the strengths, such as the problem definition is clear, the problem studied in the paper is important and critical, the paper is easy to follow, and comprehensive experiments across three distinct use cases demonstrating the effectiveness of the method. Although some concerns have been raised by reviewers, such as the experimental model is small scale, the sensitivity of the hyperparameters, concerns on greedy alignment on long sequence, the author address these problems to some extent. Overall, it is a nice paper, and we recommend an acceptance to this paper.